

# Heat and charge transport in interacting nanoconductors driven by time-modulated temperatures

Rosa López[1], Pascal Simon[2] and Minchul Lee[3*]

**1** Institut de Física Interdisciplinària i de Sistemes Complexos IFISC (CSIC-UIB),
E-07122 Palma de Mallorca, Spain
**2** Université Paris-Saclay, CNRS, Laboratoire de Physique des Solides, 91405, Orsay, France
**3** Department of Applied Physics and Institute of Natural Science, College of Applied Science,
Kyung Hee University, Yongin 17104, Korea

⋆ minchul.lee@khu.ac.kr

## Abstract

We investigate the quantum transport of the heat and the charge through a quantum dot coupled to fermionic contacts under the influence of time modulation of temperatures. We derive, within the nonequilibrium Keldysh Green's function formalism, generic formulas for the charge and heat currents by extending the concept of gravitational field introduced by Luttinger to the dynamically driven system and by identifying the correct form of dynamical contact energy. In linear response regime our formalism is validated from satisfying the Onsager reciprocity relations and demonstrates its utility to reveal nontrivial dynamical effects of the Coulomb interaction on charge and energy relaxations.



# 1  Introduction

Nanoelectronics [1] and the emerging field of Thermotronics [2–5] are at the forefront of manipulating electron charge and energy fluxes through electrostatic and thermal gradients, respectively. The interplay between these technologies, known as Thermoelectricity [6] explores how a thermal gradient influences charge current and vice versa, expanding the functionality of quantum conductors. Investigating time-dependent transport in nanostructures unlocks unique insights not achievable with static fields [7–9]. The ability to control these nanoscale systems with electrical or thermal drivings paves the way for new possibilities in quantum technologies. Electrically modulated quantum conductors lead to innovative devices like electron pumps [10–15], dynamical Coulomb Blockade quantum systems [16,17], and AC-driven nano-electromechanical systems for sensors [18]. This setup can function as a quantized emitter, operating as a quantum capacitor in the adiabatic regime and behaving like an RC circuit in the GHz range, with a unique charge relaxation resistance quantized as $R_0 = h/2e^2$ [7,19,20]. They have demonstrated to serve as single electron sources for quantum computing applications and metrology [12,21–26].

Similarly, thermally modulated nanoconductors, within the framework of quantum thermodynamics [27], are rapidly advancing. Examples include quantum thermal machines [26, 28–32], thermal diodes [33,34], thermal transistors [35–38], thermal memristors [39] and thermal capacitors [2–4]. Despite significant progress, time-dependent transport in nanostructures, especially in thermally driven systems, remains a challenging and vibrant field. Thus far, researchers have primarily tackled this issue in two scenarios: the incoherent transport regime [40] and the adiabatic driving [41–44], characterized by a slow time-modulation leading to a net transport of charge or heat. Adiabatic pumping for the design of thermal machines [45, 46] is rooted in geometrical concepts like the Berry connection and it has very broad applications in electronics, thermotronics [5], and quantum information processing. Recent studies on temperature-driven dynamics in two-level systems engineer an effective temperature through oscillator frequency manipulation [47]. However, most of the existing temperature-driven studies heavily rely on scattering theory, lacking the consideration of Coulomb interaction beyond mean-field treatment and being unable to capture dynamical excitations induced by nonadiabatic temperature driving in the presence of nontrivial interactions.

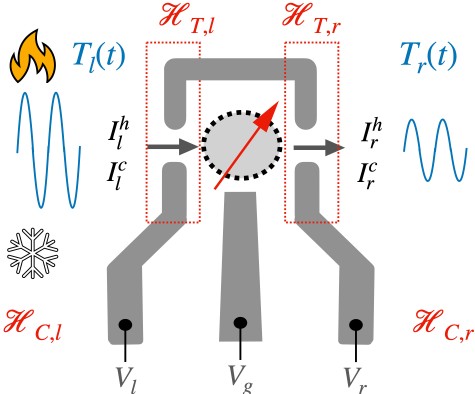

Figure 1: Lateral quantum dot system coupled to a left reservoir ($l$) and a right reservoir ($r$) that are described by $\mathcal{H}_{C,l}$ and $\mathcal{H}_{C,r}$. Each reservoir is under the influence of a modulated temperature in time with $T_l(t)$ and $T_r(t)$. Left and right tunneling barriers are described by $\mathcal{H}_{T,l}$ and $\mathcal{H}_{T,r}$, respectively, as indicated. The central part corresponds a spinful quantum dot. Plunger gates $V_l$ and $V_r$ control the barrier transparency, an additional gate $V_g$ is applied to the quantum dot region to tune the dot level position denoted by $\epsilon_\sigma$.

New and exciting functionalities are expected to arise in time-dependent thermally-driven interacting nanoconductors away from the adiabatic regime. This issue is the central objective of our work. The primary challenge in introducing a thermal bias lies in achieving a microscopic formulation. In this context, reconciling the macroscopic nature of temperature gradients and thermal forces, which arise from statistical averaging, appears to be incompatible. Then, representing these effects in a microscopic quantum mechanical Hamiltonian is not straightforward. However, in 1964, Luttinger [48,49] presented a clever and ingenious solution. He introduced a scalar potential, $\Psi$, referred as a *gravitational field*, which couples to the energy density of the system, $J_E$. This scalar field can be viewed as a mechanical conjugate to the energy density of the system defined as follows

$$\mathcal{H}_G = \int J_E(\boldsymbol{r})\Psi(\boldsymbol{r})d\boldsymbol{r}. \tag{1}$$

Luttinger justified his trick by the request for this scalar field to satisfy the Einstein relation, i.e., the potential adjusts itself to balance the thermal force, resulting in an identity $\nabla\Psi = \nabla T/T$ in the thermal equilibrium. In this respect, the gravitational field or thermomechanical potential serves as a local proxy for local temperature variations. This theoretical treatment finds application in determining linear responses to static gradients by introducing the scalar field [50,51] or equivalently, a vector potential [52]. It proves effective in various scenarios, including classical systems [53], thermoelectrical transport in quantum systems using density-functional theory framework [50] and transient current calculations [51,54] in the stationary regime. In the latter, the adiabatic limit for time-dependent temperature is analyzed using the nonequilibrium Keldysh Green function (NEGF) formalism [45,46,49,55], employing Luttinger's trick.

Our work applies Luttinger's idea to a quantum conductor tunnel-coupled to two electronic reservoirs subjected to time-dependent temperature modulation [see Fig. 1]. We focus on a quantum dot (QD), the most generic quantum conductor, featuring multiple local levels with diverse interactions such as Coulomb interaction and spin-orbit coupling. Our goal is to derive generic expressions of the electric and heat currents flowing through the QD junctions

in the framework of the NEGF formalism. For that purpose we adopt the gravitational field description to address the temperature modulation in time. Our formalism to be proposed is quite widely applicable in that it can consider diverse and complex quantum conductors and, in particular, it allows one to deal with the effect of strong Coulomb interaction in a systematic way, in contrast to previous methods. Also, it can be easily extended to involve geometric configurations such as quantum mechanical interferometers.

Our formalism, in the course of its development, addresses a very important technical aspect regarding the correct definition of the dynamical contact energy and its coupling to the gravitational field. We adopt a tight-binding model to describe the system, which divides the total Hamiltonian into pieces, namely, the contact (reservoir) regions, the quantum dot, and the tunneling barriers connecting the contacts to the QD. The contact energy responsible for the heat current in the corresponding contact

$$\mathcal{Q}_\ell = \mathcal{H}_{\text{C}\ell} + \lambda \mathcal{H}_{\text{T}\ell} \quad \text{with} \quad \lambda = 1/2, \tag{2}$$

should then include not only the energy stored in the contact $\ell = l, r$ (described by the Hamiltonian $\mathcal{H}_{\text{C}\ell}$, see Eq. (3)) but also the half of the energy stored in the tunneling barrier coupled to that contact (described by the Hamiltonian $\mathcal{H}_{\text{T}\ell}$, see Eq. (5)) [56, 57]. Together with additional physical requirements (to be explained in Sec. 3) it is natural to couple the Luttinger field to the contact energy $\mathcal{Q}_\ell$ (which therefore includes $\mathcal{H}_{\text{T}\ell}$). This is the most important ingredient for a correct application of the Luttinger's trick to the calculation of the dynamical heat current.

Accommodating the Luttinger's trick and employing the NEGF technique, we formulate expressions for charge and heat currents with a particular emphasis on the linear response regime. Notably, our formalism yields comprehensive expressions for the currents in relation to the QD NEGFs [see Eqs. (34) and (37)]. In special conditions, by a help of the charge conservation and a sum rule on the energy change rates, the currents can be obtained solely in terms of the retarded and advanced components of the QD NEGFs [see Eq. (41)], which streamlines the computation of the currents. It should be noted that our expressions for currents are the analogue to those for electrical current induced by a time-dependent electrical modulation originally presented in the pioneering work by Jauho, Wingreen and Meir [8], but in our case the time-dependent modulation is done thermally instead.

In demonstrating our Luttinger formalism, we initially apply it to the noninteracting case in which the fulfillment of the Onsager reciprocity relation validates our approach and moreover our results align with those obtained from the scattering theory. Extending our formalism to the interacting case within the Hartree approximation reveals that the Coulomb interaction can alter the responses for charging and energy relaxations with distinct temperature dependences. The success of our formalism in these applications suggests its potential for studying dynamical heat transport in the presence of the Coulomb blockade effect or many-body correlations.

Our paper is organized as follows: In Sec. 2 we introduce our model Hamiltonian and implement the Luttinger's trick onto it. We also define the charge and heat currents and find out the relevant sum rules. In Sec. 3 we express the charge and heat currents in terms of QD NEGFs by solving the corresponding Dyson's equations for the NEGF. Later, we restrict ourselves to the linear response regime and further simplify the expressions for the charge and heat currents with a help of sum rules, which are the main results of our work. Sections 4 and 5 demonstrate the applications of our Luttinger formalism, considering the noninteracting and interacting cases, respectively. In Sec. 6, we summarize our work and discuss the possible applications and extensions of our formalism and the experimental proposals.

## 2 Thermal fields and currents

For our study, we consider a quantum dot coupled to two fermionic reservoirs. To represent this configuration, we establish the Hamiltonian in the context of a tight-binding model. To construct this Hamiltonian, we first consider the distinct components that comprise our nanoconductor, and then provide an explanation for how to add the term corresponding to the gravitational field. The tight-binding Hamiltonian comprises three contributions. First, the Hamiltonian for the electrodes, denoted as $\ell$ that takes the values of left ($l$) and right ($r$), is given by

$$\mathcal{H}_\mathrm{C} = \sum_\ell \mathcal{H}_{\mathrm{C}\ell} = \sum_\ell \sum_{\mathbf{k}\sigma} \epsilon_{\ell\mathbf{k}} c^\dagger_{\ell\mathbf{k}\sigma} c_{\ell\mathbf{k}\sigma}\,, \tag{3}$$

where $c^\dagger_{\ell\mathbf{k}\sigma}$ is the creation operator for an electron in the lead $\ell$ with wavevector $\mathbf{k}$, spin $\sigma(=\uparrow,\downarrow)$, and energy $\epsilon_{\ell\mathbf{k}}$ measured with respect to the Fermi level. Second, the central conductor considered as an interacting localized multi-level quantum dot is described as

$$\mathcal{H}_\mathrm{D} = \mathcal{H}_\mathrm{D}(\{d_m, d^\dagger_m\})\,. \tag{4}$$

Here $d_m$ annihilates an electron on the localized level $m$ ($m = 1,\cdots,N_d$) where the index $m$ denotes both the orbital level and the spin. $\mathcal{H}_\mathrm{D}$ can describe any kind of interactions inside the quantum dot such as the Coulomb interactions and the spin-orbit couplings. Thirdly, the tunneling Hamiltonian that connects each of the two electrodes with the central conductor reads

$$\mathcal{H}_\mathrm{T} = \sum_\ell \mathcal{H}_{\mathrm{T}\ell} = \sum_\ell \sum_{m\mathbf{k}\sigma} \left( t_{\ell\mathbf{k}\sigma,m} d^\dagger_m c_{\ell\mathbf{k}\sigma} + t^*_{\ell\mathbf{k}\sigma,m} c^\dagger_{\ell\mathbf{k}\sigma} d_m \right)\,, \tag{5}$$

where the tunneling amplitude between the lead $\ell$ and the dot level $m$ is denoted by $t_{\ell\mathbf{k}\sigma,m}$. Hereafter, for simplicity, we assume that the tunneling amplitude is momentum-independent $t_{\ell\mathbf{k}\sigma,m} \equiv t_{\ell\sigma,m}$. Accordingly, we define the escaping tunneling rates $\Gamma_{\ell,mm'} \equiv \sum_\sigma \pi\rho_0 t_{\ell\sigma,m} t^*_{\ell\sigma,m'}/\hbar$ for level $m$ and $m'$ due to the coupling to the contact $\ell$, where $\rho_0$ is the density of states in the contacts at the Fermi energy. Later it is convenient to express the rates in the form of a $N_d \times N_d$ matrix $\mathbf{\Gamma}_\ell$ whose matrix elements are given by $\Gamma_{\ell,mm'}$.

### 2.1 Luttinger's trick and Hamiltonian

The idea about the Luttinger's trick consists of introducing new fields, dubbed as *gravitational fields* $\Psi_\ell(t)$ which are coupled to the contact energies. We determine the precise form of this coupling based on the following *two* arguments. First, as mentioned before, the contact energies in the framework of a tight-binding formulation should be redefined to account for the reactance or energy stored in the barrier [56], leading to $\mathcal{Q}_\ell$ as specified in Eq. (2). Therefore, in our setup, the term responsible for the dynamical thermal driving is introduced as

$$\mathcal{H}_\mathrm{G} = \sum_\ell \Psi_\ell(t) \left( \mathcal{Q}_\ell - \langle \mathcal{Q}_\ell \rangle_0 \right)\,, \tag{6}$$

where each of the $\Psi_\ell(t)$ field is coupled to the *excess* energy for the contact $\ell$ with respect to its equilibrium value (at $\Psi_\ell = 0$); $\langle\cdots\rangle_0$ denotes the expectation value at equilibrium. The coupling to the excess energy is reasonable because for sufficiently weak driving or at low temperature only the excitations around the Fermi level will contribute to the electric and thermal transports. Furthermore, mathematically, it introduces only the additional time-varying number, $-\sum_\ell \Psi_\ell(t) \langle \mathcal{Q}_\ell \rangle_0$ which does not affect the dynamics of states. The constant $\lambda$ in $\mathcal{Q}_\ell$, the

coupling coefficient of the gravitational field to the tunneling barrier energy, is set to be 1/2, but for a time being we keep this symbol as it is in order to track down how the precise value of this coefficient influences the results.

Second, we require that the coupling to the gravitational field should not affect the effective coupling between the dot and the leads [49, 55] since the dot-lead coupling should be immune to the temperature in the leads. Interestingly, we find that the coupling in Eq. (6) automatically satisfies this requirement, at least, in the linear response regime, as we will see later [see Sec. 3.1].

The complete Hamiltonian for our setup is then described by

$$\mathcal{H}(t) = \mathcal{H}_{\mathrm{C}} + \mathcal{H}_{\mathrm{D}} + \mathcal{H}_{\mathrm{T}} + \mathcal{H}_{\mathrm{G}}(t) = \mathcal{H}_{\mathrm{C},\Psi}(t) + \mathcal{H}_{\mathrm{D}} + \mathcal{H}_{\mathrm{T},\Psi}(t), \tag{7}$$

where we have introduced the time-dependent contact and tunneling Hamiltonians defined as

$$\mathcal{H}_{\alpha,\Psi}(t) \equiv \sum_{\ell} \mathcal{H}_{\alpha\ell,\Psi}(t) = \sum_{\ell} \left[ \mathcal{H}_{\alpha\ell} + \lambda_{\alpha} \Psi_{\ell}(t)(\mathcal{H}_{\alpha\ell} - \langle \mathcal{H}_{\alpha\ell} \rangle_0) \right], \tag{8}$$

for $\alpha = \mathrm{C}, \mathrm{T}$ and with $\lambda_{\mathrm{C}} = 1$ and $\lambda_{\mathrm{T}} = \lambda$. We assume that the thermal drivings on both the contacts are periodic with the same frequency $\Omega = 2\pi/\tau$: $\Psi_{\ell}(t+\tau) = \Psi_{\ell}(t)$. More specifically, we take a sinusoidal time dependence:

$$\Psi_{\ell}(t) = \Psi_{\ell} \cos \Omega t. \tag{9}$$

Here the factors $\Psi_{\ell}$ are real and can be zero. Since the dynamical variation of the temperature is taken into account via the gravitaional field, both the contacts are assumed to have the same chemical potential and the same base temperature $T$ (or the inverse temperature $\beta$) so that the thermal populations of both the uncoupled contacts are specified by the same Fermi distribution function $f(\epsilon)$.

## 2.2 Charge currents and charge conservation

We focus in our study on the charge and heat currents transversing the contacts, driven by the dynamical change of the temperature. The charge currents, or the change rates of the charges in the contacts ($\ell = l, r$) are described in terms of the charge current operators

$$\mathcal{I}_{\ell}^c \equiv -e \frac{d\mathcal{N}_{\ell}}{dt} = -\frac{ie}{\hbar}[\mathcal{H}, \mathcal{N}_{\ell}], \tag{10}$$

where $\mathcal{N}_{\ell} \equiv \sum_{\mathbf{k}\sigma} c_{\ell\mathbf{k}\sigma}^{\dagger} c_{\ell\mathbf{k}\sigma}$ is the charge number operator for the contact $\ell$. Under this consideration, the charge currents, the expectation values of the charge current operators for the contacts and the quantum dot can be obtained by evaluating

$$I_{\ell}^c(t) = -\frac{ie}{\hbar} \sum_{m\mathbf{k}\sigma} t_{\ell\mathbf{k}\sigma,m}(t) \langle d_m^{\dagger}(t) c_{\ell\mathbf{k}\sigma}(t) \rangle + (c.c), \tag{11a}$$

$$I_{\mathrm{D}}^c(t) = -e \frac{d\langle \mathcal{N}_{\mathrm{D}} \rangle}{dt} = -e \sum_m \frac{d\langle n_m \rangle}{dt}, \tag{11b}$$

where $\mathcal{N}_{\mathrm{D}} \equiv \sum_m n_m$ (with $n_m \equiv d_m^{\dagger} d_m$) is the charge number operator for the quantum dot. It should be noted that the charge conservation condition, $[\mathcal{H}, \sum_{\ell} \mathcal{N}_{\ell} + \mathcal{N}_{\mathrm{D}}] = 0$, guarantees that their sum vanishes at all time $t$:

$$\sum_{\ell} I_{\ell}^c(t) + I_{\mathrm{D}}^c(t) = 0. \tag{12}$$

Later we will use this equality to simplify our final results.

## 2.3 Heat currents and sum rule

The heat currents for the electrodes are derived from the time derivative of the energies stored at the contacts. Then, according to our choice of the contact energy $\mathcal{Q}_\ell$ which incorporates the contribution from the neighboring tunneling barrier, the heat current for contact $\ell$ is given by [56,57]

$$I_\ell^h(t) = \frac{d\langle\mathcal{Q}_\ell(t)\rangle}{dt}. \tag{13}$$

It should be noted[1] that multiple choices for the definition of the heat currents for contacts are *a priori* possible in the Luttinger formalism, while we find that Eq. (13) is the most suitable one.

The power supplied by the time-dependent thermal source or the power dissipated is defined as $P(t) = \langle\partial H/\partial t\rangle = \langle\partial H_G/\partial t\rangle$ and explicitly expressed as

$$P(t) = \sum_\ell \dot\Psi_\ell \left(\langle\mathcal{Q}_\ell\rangle - \langle\mathcal{Q}_\ell\rangle_0\right), \tag{14}$$

which contains the source contributions from the contacts and the tunneling barriers.

Under the thermal driving, the energy conservation does not hold: $P(t) \neq 0$. However, one can still find an equality similar to Eq. (12). To this purpose, we define the energy change rates as

$$W_{\alpha\ell}(t) \equiv \frac{i}{\hbar}[\mathcal{H}(t), \mathcal{H}_{\alpha\ell,\Psi}(t)] \quad (\alpha = \mathrm{C}, \mathrm{T}), \tag{15a}$$

$$W_{\mathrm{D}}(t) \equiv \frac{i}{\hbar}[\mathcal{H}(t), \mathcal{H}_{\mathrm{D}}]. \tag{15b}$$

Then, from Eq. (7), the obvious commutation relation $[\mathcal{H}(t), \mathcal{H}(t)] = 0$ leads to a useful equality

$$\sum_\ell (W_{\mathrm{C}\ell}(t) + W_{\mathrm{T}\ell}(t)) + W_{\mathrm{D}}(t) = 0, \tag{16}$$

which is to be exploited later.

## 3 Charge and heat currents in terms of NEGF's

In the seminal research [8], the charge current flowing through a quantum dot under a dynamical electric drive was formulated in a closed form in terms of the interacting QD Green functions. In a similar way, we formulate here the charge and heat currents in terms of solely the QD NEGFs when the system is driven by oscillating temperatures. For such purpose we adopt the nonequilibrium Keldysh formalism and employ the equation-of-motion technique

---

[1]A potential argument could state that the appropriate selection of the contact energy operator is $\mathcal{Q}_\ell = \mathcal{H}_{\mathrm{C}\ell,\Psi}(t) + \lambda\mathcal{H}_{\mathrm{T}\ell,\Psi}(t)$, since it incorporates the sources, that is, the contribution to the energy from the coupling terms of the gravitional fields. Evaluating this alongside Eq. (2), we observe their difference being second order in $\Psi_\ell$. As we are operating in a linear response regime, both definitions offer equivalent outcomes for the heat current to this order. The determination of the correct contact energy operator definition requires a study into the nonlinear regime, which currently surpasses our research scope. Besides, the power definition is of second order in $\Psi_\ell$, which means that the source contribution to the heat fluxes is of second order and it is consistent with the indistinguishability between the two candidate definitions for the contact energy.

together with the Langreth rules [58]. Throughout our paper, we use the retarded/advanced and lesser QD NEGFs defined as

$$\mathcal{G}_{mm'}^{R/A}(t,t') \equiv \mp i\Theta(\pm(t-t'))\langle\{d_m(t), d_{m'}^\dagger(t')\}\rangle, \tag{17a}$$

$$\mathcal{G}_{mm'}^{<}(t,t') \equiv i\langle d_{m'}^\dagger(t')d_m(t)\rangle, \tag{17b}$$

respectively, and the contact and QD-contact NEGFs are defined in a similar way: For examples,

$$\mathcal{G}_{\ell\mathbf{k}\sigma,\ell'\mathbf{k}'\sigma'}^{<}(t,t') \equiv i\langle c_{\ell'\mathbf{k}'\sigma'}^\dagger(t')c_{\ell\mathbf{k}\sigma}(t)\rangle, \tag{18a}$$

$$\mathcal{G}_{m,\ell\mathbf{k}\sigma}^{<}(t,t') \equiv i\langle c_{\ell\mathbf{k}\sigma}^\dagger(t')d_m(t)\rangle. \tag{18b}$$

Based on the definition of the Green's functions, the charge currents (11) and the expectation values for the contact energy operators (2) are readily expressed in terms of the NEGF´s:

$$I_\ell^c(t) = e\sum_{m\mathbf{k}\sigma} \frac{t_{\ell\mathbf{k}\sigma,m}^*(t)}{\hbar}\mathcal{G}_{m,\ell\mathbf{k}\sigma}^{<}(t,t) + (c.c.), \tag{19a}$$

$$I_{\mathrm{D}}^c(t) = e\sum_{\sigma} i\frac{d}{dt}\mathcal{G}_{mm}^{<}(t,t), \tag{19b}$$

and

$$\langle\mathcal{H}_{\mathrm{C}\ell}\rangle = -i\sum_{\mathbf{k}\sigma}\epsilon_{\ell\mathbf{k}}\mathcal{G}_{\ell\mathbf{k}\sigma,\ell\mathbf{k}\sigma}^{<}(t,t), \tag{20a}$$

$$\langle\mathcal{H}_{\mathrm{T}\ell}\rangle = -i\sum_{m\mathbf{k}\sigma}t_{\ell\mathbf{k}\sigma,m}^*\mathcal{G}_{m,\ell\mathbf{k}\sigma}^{<}(t,t) + (c.c.). \tag{20b}$$

Note that above we have introduced a time-varying effective tunneling amplitudes,

$$t_{\ell\mathbf{k}\sigma,m}(t) \equiv (1 + \lambda\Psi_\ell(t))t_{\ell\mathbf{k}\sigma,m}, \tag{21}$$

for convenience. Knowing the time dependence of two expectation values, $\langle\mathcal{H}_{\mathrm{C}\ell}\rangle$ and $\langle\mathcal{H}_{\mathrm{T}\ell}\rangle$, one can evaluate not only the heat currents (13) and the power (14) but also the energy change rates for contact $\ell$ and the tunneling parts via

$$W_{\alpha\ell}(t) = (1 + \lambda_\alpha\Psi_\ell(t))\frac{d\langle\mathcal{H}_{\alpha\ell}\rangle}{dt} \quad (\alpha = \mathrm{C}, \mathrm{T}). \tag{22}$$

Now, by employing the equation-of-motion technique [8], the lesser Green's functions, $\mathcal{G}_{m,\ell\mathbf{k}\sigma}^{<}(t,t)$ and $\mathcal{G}_{\ell\mathbf{k}\sigma,\ell\mathbf{k}\sigma}^{<}(t,t)$, can be cast in terms of solely the QD NEGFs [see Appendix A for details]. After some algebraic manipulations on $\mathcal{G}_{m,\ell\mathbf{k}\sigma}^{<}(t,t)$, the contact charge current (19a) and the expectation value of energy stored in the tunneling barrier (20b) have compact forms in terms of self energies and the QD NEGFs:

$$I_\ell^c(t) = e\int dt' \mathrm{Tr}\big[\mathbf{G}^R(t,t')\mathbf{\Sigma}_\ell^{<}(t',t) + \mathbf{G}^{<}(t,t')\mathbf{\Sigma}_\ell^A(t',t)\big] + (c.c.), \tag{23a}$$

$$\langle\mathcal{H}_{\mathrm{T}\ell}\rangle = -\frac{i\hbar}{1+\lambda\Psi_\ell(t)}\int dt' \mathrm{Tr}\big[\mathbf{G}^R(t,t')\mathbf{\Sigma}_\ell^{<}(t',t) + \mathbf{G}^{<}(t,t')\mathbf{\Sigma}_\ell^A(t',t)\big] + (c.c.). \tag{23b}$$

Here $\mathbf{G}^a$ and $\mathbf{\Sigma}^a$ ($a = R, A, <$) are the $N_d \times N_d$ matrix representations of the QD Green's functions and the self energies, respectively. The traces are done over the QD orbital degrees of freedoms. The matrix elements of the self energies are defined as

$$\Sigma_{\ell,mm'}^a(t,t') \equiv \sum_{\mathbf{k}\sigma} \frac{t_{\ell\mathbf{k}\sigma,m}(t)}{\hbar}g_{\ell\mathbf{k}\sigma}^a(t,t')\frac{t_{\ell\mathbf{k}\sigma,m'}^*(t')}{\hbar}, \tag{24}$$

where $g_{\ell\mathbf{k}\sigma}^{a}(t,t')$ with $a = R, A, <$ correspond to the *uncoupled* contact Green's functions governed by the time-dependent Hamiltonian, $\epsilon_{\ell\mathbf{k}}(t)c_{\ell\mathbf{k}\sigma}^{\dagger}c_{\ell\mathbf{k}\sigma}$, with the time-varying energy,

$$\epsilon_{\ell\mathbf{k}}(t) \equiv (1 + \Psi_\ell(t))\epsilon_{\ell\mathbf{k}}. \tag{25}$$

It should be noted that the gravitational field $\Psi_\ell(t)$ enters into the self energy through the time-varying tunneling amplitude $t_{\ell\mathbf{k}\sigma,m}(t)$ as well as the dynamically driven contact energy $\epsilon_{\ell\mathbf{k}}(t)$.

Similarly, by expressing $\mathcal{G}_{\ell\mathbf{k}\sigma,\ell\mathbf{k}\sigma}^{<}(t,t)$ in terms of the QD NEGFs, the expectation value of the contact Hamiltonian is written as

$$\langle \mathcal{H}_{C\ell} \rangle - E_{C\ell 0} = \int dt' \int dt'' \, \mathrm{Tr}\big[\boldsymbol{\Xi}_\ell^{AR}(t,t'',t')\mathbf{G}^<(t',t'') \tag{26}$$
$$+ \boldsymbol{\Xi}_\ell^{<R}(t,t'',t')\mathbf{G}^R(t',t'') + \boldsymbol{\Xi}_\ell^{A<}(t,t'',t')\mathbf{G}^A(t',t'')\big],$$

where $E_{C\ell 0} \equiv -i \sum_{\mathbf{k}\sigma} \epsilon_{\ell\mathbf{k}} g_{\ell\mathbf{k}\sigma}^{<}(t,t)$ are the *equilibrium* energies contained in the *unperturbed* contacts. Note that this constant is irrelevant in our study of heat flow because it does not contribute to the heat current once the time derivative is taken. The matrix elements of the self-energy-like term $\boldsymbol{\Xi}_\ell^{ab}$ in Eq. (26) are defined as

$$\Xi_{\ell,m'm''}^{ab}(t,t',t'') \equiv -i \sum_{\mathbf{k}\sigma} \frac{t_{\ell\mathbf{k}\sigma,m'}(t')}{\hbar} g_{\ell\mathbf{k}\sigma}^{a}(t',t)\epsilon_{\ell\mathbf{k}}g_{\ell\mathbf{k}\sigma}^{b}(t,t'')\frac{t_{\ell\mathbf{k}\sigma,m''}^{*}(t'')}{\hbar}, \tag{27}$$

with $a, b = R, A, <$.

In the presence of the time-dependent terms in the Hamiltonian, the Green's functions $\mathcal{G}(t,t')$ depend not on the time difference $t - t'$ but on $t$ and $t'$ separately. However, since the driving is periodic with period $\tau$, the Green's functions and the self energies are also periodic with $\mathcal{G}(t+\tau, t'+\tau) = \mathcal{G}(t,t')$ so that it is more convenient to apply the Fourier transformation and to move from the time domain to the frequency domain. We adopt the mixed time-energy representation for the Fourier transformation:

$$\mathcal{G}(t,t') = \int_{-\infty}^{\infty} \frac{d\omega}{2\pi} e^{-i\omega(t-t')}\mathcal{G}(t,\omega) \quad \text{and} \quad \mathcal{G}(t,\omega) = \sum_{n=-\infty}^{\infty} \mathcal{G}(n,\omega)e^{-in\Omega t}. \tag{28}$$

Then, the integrals in Eqs. (23) and (26) can be expressed in terms of the Fourier components:

$$\int dt' \, \mathrm{Tr}\big[\mathbf{G}^a(t,t')\boldsymbol{\Sigma}_\ell^b(t',t)\big] = \sum_{nn'} \int \frac{d\omega}{2\pi} e^{-i(n+n')\Omega t} \, \mathrm{Tr}\big[\mathbf{G}^a(n,\omega+n'\Omega)\boldsymbol{\Sigma}_\ell^b(n',\omega)\big], \tag{29a}$$

$$\int dt' \int dt'' \, \mathrm{Tr}\big[\boldsymbol{\Xi}_\ell^{ab}(t,t'',t')\mathbf{G}^c(t',t'')\big] = \sum_{nn'} \int \frac{d\omega}{2\pi} e^{-i(n+n')\Omega t} \, \mathrm{Tr}\big[\boldsymbol{\Xi}_\ell^{ab}(n,n',\omega)\mathbf{G}^c(n',\omega)\big], \tag{29b}$$

with

$$\Xi_\ell^{ab}(n,n',\omega) \equiv \frac{1}{\tau}\int_0^\tau dt \int dt' \int dt'' \Xi_\ell^{ab}(t,t'',t')e^{-i\omega(t'-t'')}e^{-in'\Omega(t'-t)}e^{in\Omega t}. \tag{30}$$

Importantly, the charge and heat currents [see Eqs. (23) and (26) together with Eq. (13)] are now expressed in the frequency domain employing Eqs. (29a) and (29b) *solely* in terms of the QD NEGF. This constitutes an *exact* form for the charge and heat currents of an interacting conductor coupled to fermionic reservoirs.

One may want to interpret the above ac-driven current in terms of the photo-assisted tunneling as done in the Tien-Gordon approach [59]. In this approach, the effect of the driving is transformed to the appearance of the quasi-energy states $(\omega + n\Omega)$ due to the absorption/emission processes of photons and the resulting current is then given by the sum of all possible processes, each of which is weighted by proper Bessel functions with argument proportional to $\Psi_\ell$. This simple interpretation does not work in our case though. The main reason is that, while in the Tien-Gordon approach the time-dependent field is coupled to the charge numbers $(c^\dagger_{\ell\mathbf{k}\sigma} c_{\ell\mathbf{k}\sigma})$ only, in the temperature-driven case it is coupled to the excitation energies $(\epsilon_{\ell\mathbf{k}})$ as well. Then, the photon-assisted processes with different $n$ are no longer independent of each other but are instead intermingled with each other and have very complicated energy dependencies [see Eq. (84)]. Hence, it is not practical to interpret the temperature-driven current in terms of the photon-assisted processes.

Hereafter we adopt an approximation scheme using a *linear expansion in the Luttinger field* $\Psi_\ell(t)$. The reasons are three-fold: (1) The Luttinger scheme was originally proposed to work only in the linear response regime, (2) the second requirement of our Luttinger setup, making the effective dot-contact coupling independent of $\Psi_\ell$, is satisfied only in the linear response regime, and (3) in a practical manner the use of the above exact form is limited because it is quite difficult to obtain manageable analytical expressions for $\Sigma^a_\ell(n,\omega)$ and $\Xi^{ab}_\ell(n,n',\omega)$ working for arbitrary magnitude of $\Psi_\ell$.

## 3.1 Linear response regime

After deriving the general forms for the charge and heat currents in the system, our intention is to supply a manageable formulation of such currents in terms of the equilibrium or dynamical QD Green's functions. This objective is reachable within the linear response regime, in which we treat the amplitudes $\Psi_\ell$ as the smallest parameters and keep up to their *linear order* in the charge and heat fluxes.

Considering that our periodic driving (9) is of the form, $\Psi_\ell(t) = (\Psi_\ell/2)(e^{i\Omega t} + e^{-i\Omega t})$, in the linear expansion with respect to $\Psi_\ell$, only the Fourier components with $n, n' = 0, \pm 1$ of $\Sigma^a_\ell(n,\omega)$ and $\Xi^{ab}_\ell(n,n',\omega)$ should remain finite. While their explicit expressions and derivations can be found in Appendix B, one should note that from Eq. (83) the linear expansion of $\Sigma^{R/A}_{\ell\sigma}(t,t')$ is given by

$$\Sigma^{R/A}_\ell(t,t') \approx \mp i\mathbf{\Gamma}_\ell \left(1 + (2\lambda - 1)\Psi_\ell(t)\right)\delta(t-t'), \tag{31}$$

so that $\Sigma^{R/A}_\ell(t,t')$ becomes independent of the gravitational field only when $\lambda = 1/2$. That is, only at this choice of $\lambda$, the requirement that the dot-contact hybridization should be immune to the temperature change is met.

In the same spirit of the linear expansion, only the $n = 0, \pm 1$ Fourier components of the QD Green's functions should survive: $\mathbf{G}^a(0,\omega)$ is in the zeroth order of $\Psi_\ell$, while $\mathbf{G}^a(\pm 1,\omega)$ are linear in $\Psi_\ell$: up to the linear order in $\Psi_\ell$,

$$\mathbf{G}^a(t,\omega) \approx \mathbf{G}^a(0,\omega) + e^{-i\Omega t}\mathbf{G}^a(1,\omega) + e^{+i\Omega t}\mathbf{G}^a(-1,\omega). \tag{32}$$

In particular, the $n = 0$ components of the self energies and the QD Green's functions should exactly correspond to their equilibrium values at $\Psi_\ell = 0$.

In the linear response regime, the charge and heat currents are described by their first Fourier components only,

$$I^{c/h}_{\ell/D}(t) = I^{c/h}_{\ell/D}(\Omega)e^{-i\Omega t} + I^{c/h}_{\ell/D}(-\Omega)e^{+i\Omega t}, \tag{33}$$

with $I_{\ell/\mathrm{D}}^{c/h}(-\Omega) = [I_{\ell/\mathrm{D}}^{c/h}(\Omega)]^*$ since no current flows at $\Psi_\ell = 0$. In the following sections, we are going to express the Fourier components $I_\ell^{c/h}(\Omega)$ of the charge and heat currents in terms of the equilibrium QD Green's functions, $\mathbf{G}^a(0, \omega)$ and the nonequilibrium linear components, $\mathbf{G}^a(\pm 1, \omega)$.

## 3.2 Charge/heat currents and power

In order to obtain the charge current in the linear response regime, we express the contact charge current (23a) in terms of the Fourier components $\mathbf{G}^a(n, \omega)$ and $\Sigma_\ell^a(n, \omega)$ by using Eq. (29a) and then keep only the linear-order terms in $\Psi_\ell$ [refer to Appendix B]. The derivation is quite straightforward and we obtain

$$
\begin{aligned}
I_\ell^c(\Omega) = e \int \frac{d\omega}{2\pi} \mathrm{Tr}\Bigg[ &(2i\mathbf{\Gamma}_\ell)\Bigg( -\frac{\Psi_\ell}{2}\Delta_f(\omega+\Omega,\omega)\Big(\omega+\frac{\Omega}{2}\Big)\big(\mathbf{G}^R(0,\omega+\Omega)-\mathbf{G}^A(0,\omega)\big) \\
&+ f(\omega+\Omega)\big(\mathbf{G}^R(1,\omega+\Omega)-\mathbf{G}^A(1,\omega)\big)+\mathbf{G}^<(1,\omega)\Bigg)\Bigg],
\end{aligned}
\tag{34}
$$

and

$$
I_\mathrm{D}^c(\Omega) = e \int \frac{d\omega}{2\pi} \Omega\, \mathrm{Tr}\, \mathbf{G}^<(1,\omega),
\tag{35}
$$

where we have defined

$$
\Delta_f(\omega, \omega') \equiv \frac{f(\omega)-f(\omega')}{\omega-\omega'},
\tag{36}
$$

[refer to Appendix C for details]. As expected, the charge current depends not only on the equilibrium QD Green's functions but also on the dynamical ones, $\mathbf{G}^{R/A/<}(1,\omega)$, even though the linear response ($\Psi_\ell \to 0$) is taken. It is obviously because our perturbations are dynamical and the dynamical excitations of the system, even though being small, cannot be described solely in terms of the equilibrium Green's functions.

Now we turn to the heat transport. In order to find the expressions for the heat currents, the power, and the energy change rates in the linear response regime, we write down the energies (23b) and (26) in terms of the Fourier components of $\mathbf{G}^a(n, \omega)$, $\Sigma_\ell^a(n, \omega)$, and $\Xi_\ell^{ab}(n, n', \omega)$ by using Eqs. (29a) and (29b) and then keep only the linear-order terms in $\Psi_\ell$ [see Appendix B for details]. Following the explicit derivation in Appendix C, the contact heat current can be explicitly obtained as

$$
\begin{aligned}
I_\ell^h(\Omega) = \hbar \int \frac{d\omega}{2\pi} \mathrm{Tr}\Bigg[ &(2i\mathbf{\Gamma}_\ell)\Bigg(\frac{\Psi_\ell}{2}\Delta_f(\omega+\Omega,\omega)\Big(\omega+\frac{\Omega}{2}\Big)^2\big(\mathbf{G}^R(0,\omega+\Omega)-\mathbf{G}^A(0,\omega)\big) \\
&- \Big(\omega+\frac{\Omega}{2}\Big)\big(f(\omega)\mathbf{G}^R(1,\omega)-f(\omega+\Omega)\mathbf{G}^A(1,\omega)+\mathbf{G}^<(1,\omega)\big)\Bigg)\Bigg] + I_{\ell\mathrm{T}}^h(\Omega),
\end{aligned}
\tag{37}
$$

where $I_{\ell\mathrm{T}}^h(\Omega) \equiv -\frac{\Psi_\ell}{2}\frac{i\Omega}{4}E_{\mathrm{T}\ell 0}$ is an additional unphysical term. Equations (34) and (37) are our main results.

The expression given in (37) for the contact heat current needs a discussion. The additional term $I_{\ell\mathrm{T}}^h(\Omega)$ which is proportional to $E_{\mathrm{T}\ell 0}$ is an *artefact* of our Luttinger's trick. In our setup, we dynamically drive the contact by the field $\Psi_\ell(t)$ and the tunneling barrier by the field $\lambda\Psi_\ell(t)$ so that an effective energy capacitor which is dynamically driven by the field difference $(1-\lambda)\Psi_\ell(t)$ is formed. However, this effect is not contained in the original system and is solely due to the Luttinger's setup itself. This artificial setup then gives rise to an additional heat transfer $(1-\lambda)\Psi_\ell(t)E_{\mathrm{T}\ell 0}$ (up to the linear order) between the contact $\ell$ and the tunneling

barrier, resulting in $I^h_{\ell\mathrm{T}}(\Omega)$ in the contact heat current. In fact, in the next section for the noninteracting system, we compare the results from our Luttinger's trick and those obtained from the scattering theory based on the dissipation-fluctuation theorem and find that two results are identical only when this artefact is not taken into account. Therefore, we will drop out the term $I^h_{\ell\mathrm{T}}(\Omega)$ from now on.

Finally, we find the expression for the power of dissipation (14) in the linear response regime [see Eq. (101)]. In particular, we focus on the time average of the power which is simplified to

$$\overline{P} = -\sum_{\ell} \Psi_{\ell} \, \mathrm{Re}[I^h_{\ell}(\Omega)]. \tag{38}$$

As expected, it reflects that the time-averaged power is directly related to the real part of the Fourier component of the contact heat currents. It is because the dissipation happens at the contacts and the time average picks up only the dissipative effect.

### 3.3 Application of conservation law and sum rule

Note that our expressions for the charge and heat currents, Eqs. (34) and (37) necessitate the knowledge of the nonequilibrium components $\mathbf{G}^{<}(1,\omega)$ as wells as $\mathbf{G}^{R/A}(1,\omega)$. Technically, it is much harder to theoretically obtain the lesser Green's functions than the retarded/advanced Green's functions because the lesser ones reflect not only the energy excitations of the system but also their nonequilibrium distribution. Therefore, our expressions for the currents would be more useful if the knowledge of the lesser Green's functions is avoided, especially when the quantum dot is interacting.

We have found that it is possible as long as the hybridization matrix $\boldsymbol{\Gamma}_{\ell}$ is proportional to the identity matrix:

$$\boldsymbol{\Gamma}_{\ell} = \Gamma_{\ell}\mathbf{1}, \tag{39}$$

in which condition, $\mathrm{Tr}[\boldsymbol{\Gamma}_{\ell}\mathbf{G}^{<}(1,\omega)] = \Gamma_{\ell}\,\mathrm{Tr}[\mathbf{G}^{<}(1,\omega)]$. This condition cannot accommodate the general situations in real experiments, but most of qualitative features can be still captured within this condition. So this simplifying condition is acceptable at least for theoretical studies.

First, the expression for the contact charge current, Eq. (34) can be simplified further if the charge conservation is taken into account. If we apply the charge conservation (12) up to the linear order, then we have

$$\sum_{\ell} I^c_{\ell}(\Omega) + I^c_d(\Omega) = 0, \tag{40}$$

which enables one to write down the integral $\int d\omega \, \mathrm{Tr}[\mathbf{G}^{<}(1,\omega)]$ in terms of the other components of the QD Green's functions [see Eq. (93)]. Then, we can get a nice expression for the *interacting* charge current at the contact $\ell$ in the linear response regime which writes *solely* in terms of the retarded/advanced QD Green's functions:

$$
\begin{aligned}
I^c_{\ell}(\Omega) = e \int \frac{d\omega}{2\pi} \Bigg[ & \left( \sum_{\ell'} \frac{\Psi_{\ell'}}{2} \frac{2i\Gamma_{\ell'}}{2i\Gamma + \Omega} - \frac{\Psi_{\ell}}{2} \right) \\
& \times \Delta_f(\omega+\Omega,\omega) \left( \omega + \frac{\Omega}{2} \right) (2i\Gamma_{\ell}) \, \mathrm{Tr}\left[ \mathbf{G}^R(0,\omega+\Omega) - \mathbf{G}^A(0,\omega) \right] \\
& + \frac{\Omega}{2i\Gamma + \Omega} f(\omega+\Omega)(2i\Gamma_{\ell}) \, \mathrm{Tr}\left[ \mathbf{G}^R(1,\omega+\Omega) - \mathbf{G}^A(1,\omega) \right] \Bigg],
\end{aligned}
\tag{41}
$$

with

$$\Gamma \equiv \sum_{\ell} \Gamma_{\ell} \,. \tag{42}$$

The expression of the contact heat current, Eq. (37) can be also further simplified in a similar way. Recall that we have the sum rule (16) for the energy change rates, which gives rise to

$$\sum_{\ell} (W_{C\ell}(\Omega) + W_{T\ell}(\Omega)) + W_D(\Omega) = 0 \,, \tag{43}$$

in the linear response regime. This sum rule enables us to replace the integral $\int d\omega\, \omega\, \mathrm{Tr}[\mathbf{G}^{<}(1,\omega)]$ with ones of other QD Green's functions. However, this sum rule cannot be constructed without knowing explicitly the form of the QD Hamiltonian $\mathcal{H}_D$ since the change rate $W_D(\Omega)$ depends on $\mathcal{H}_D$. Therefore, our strategy is as follows: Once $\mathcal{H}_D$ is known, we find out the expression for $W_D(\Omega)$ in terms of the QD NEGFs in the linear response regime and then write down the integral $\int d\omega\, \omega\, \mathrm{Tr}[\mathbf{G}^{<}(1,\omega)]$ in terms of other QD NEGFs by using the sum rule (43) and the explicit expression for the partial sum, $\sum_{\ell}(W_{C\ell}(\Omega) + W_{T\ell}(\Omega))$ [see Eq. (100) in Appendix C].

In the following sections, we apply our formalism to two specific examples: the noninteracting case and the interacting case with the Hartree approximation. Especially, in the noninteracting case, we demonstrate the justification of the choice $\lambda = 1/2$ in more details.

## 4  Noninteracting case

As a first application of our formalism derived in the previous sections, we consider the noninteracting case in which the QD Hamiltonian (4) is given by

$$\mathcal{H}_D = \sum_{\sigma} \epsilon_{\sigma} d_{\sigma}^{\dagger} d_{\sigma} \,. \tag{44}$$

We assume that the dot-lead coupling is spin-independent so that $\mathbf{\Gamma}$, now being $2 \times 2$ matrix, is diagonal and given by $\mathbf{\Gamma} = \Gamma \mathbf{1}$. The orbital index $m$ is now replaced by the spin index $\sigma$, and the trace over $m$ is now the summation over $\sigma$. It is then quite straightforward to derive and solve the Dyson's equations for the QD NEGFs, $\mathcal{G}_{\sigma}^{R/A}(t,t')$ and to find out their Fourier components in the linear response regime [refer to Appendix D for detailed derivations]. The $n = 0$ (equilibrium) components of the retarded/advanced QD Green's functions are found to be

$$\mathcal{G}_{\sigma}^{R/A}(0,\omega) = \frac{1}{\omega - \epsilon_{\sigma}/\hbar \pm i\Gamma} \,, \tag{45}$$

and their $n = 1$ components exactly vanish at $\lambda = 1/2$, that is, $\mathcal{G}_{\sigma}^{R/A}(\pm 1, \omega) = 0$ since $\Sigma_{\ell\sigma}^{R/A}(\pm 1, \omega) = 0$. On the other hand, $\mathcal{G}_{\sigma}^{<}(\pm 1, \omega)$ does not vanish at $\lambda = 1/2$, so the dynamical components of the QD Green's functions are still relevant in time-dependent charge and heat transports. However, as explained in the previous section, we do not seek out the explicit expression for $\mathcal{G}_{\sigma}^{<}(\pm 1, \omega)$, but instead resort to the charge conservation (40) and the sum rule (43) to derive the integrals of $\mathcal{G}_{\sigma}^{<}(\pm 1, \omega)$.

Starting from the general formulas of the charge and heat currents, Eqs. (41) and (37), one can derive the explicit expressions for the charge and heat currents [see Eqs. (107) and

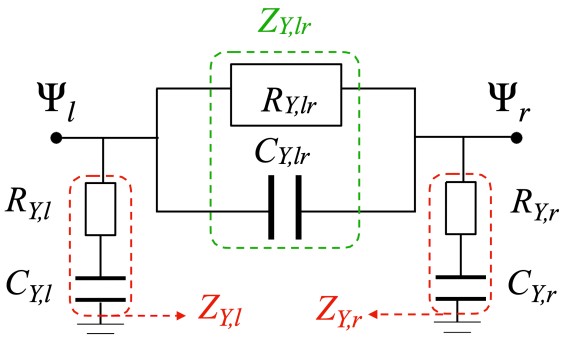

Figure 2: Equivalent RC circuit in the low temperature and low frequency limits. The RC circuit describes both the thermoelectrical (with $Y = L$) and the thermal admittance (with $Y = K$) in terms of resistors $R_{Y,\ell}$, capacitors $C_{Y,\ell}$ and the cross resistance and capacitances $R_{Y,lr}$, and $C_{Y,lr}$. Refer to the main text for their specifications for either the thermoelectrical or the thermal transports.

(113)] for the noninteracting case, in terms of the *thermoelectric admittances* $L_{\ell\ell'}(\Omega)$ and the *thermal admittances* $K_{\ell\ell'}(\Omega)$:

$$I_\ell^c(\Omega) = \sum_{\ell'} L_{\ell\ell'}(\Omega)\frac{\Psi_{\ell'}}{2}, \qquad I_\ell^h(\Omega) = \sum_{\ell'} K_{\ell\ell'}(\Omega)\frac{\Psi_{\ell'}}{2}, \qquad (46)$$

where the diagonal components are decomposed into $L_{\ell\ell}(\Omega) \equiv L_\ell(\Omega) - L_{\ell\bar{\ell}}(\Omega)$ and $K_{\ell\ell}(\Omega) \equiv K_\ell(\Omega) - K_{\ell\bar{\ell}}(\Omega)$. The self admittances are then found to be

$$L_\ell(\Omega) = (2i\Gamma_\ell)e\Omega\sum_\sigma P_{1\sigma}(\Omega), \qquad (47a)$$

$$K_\ell(\Omega) = (2i\Gamma_\ell)(-\hbar)\Omega\sum_\sigma P_{2\sigma}(\Omega), \qquad (47b)$$

and the cross admittances are given by

$$L_{\ell\bar{\ell}}(\Omega) = L_{\bar{\ell}\ell}(\Omega) = 4\Gamma_l\Gamma_r e\sum_\sigma P_{1\sigma}(\Omega), \qquad (48a)$$

$$K_{\ell\bar{\ell}}(\Omega) = K_{\bar{\ell}\ell}(\Omega) = 4\Gamma_l\Gamma_r(-\hbar)\sum_\sigma P_{2\sigma}(\Omega). \qquad (48b)$$

where we have defined

$$P_{n\sigma}(\omega) \equiv \int \frac{d\omega}{2\pi}\Delta_f(\omega+\Omega,\omega)\left(\omega+\frac{\Omega}{2}\right)^n \mathcal{G}_\sigma^R(0,\omega+\Omega)\mathcal{G}_\sigma^A(0,\omega). \qquad (49)$$

These linear-response admittances can be expressed in terms of an equivalent RC circuit, as shown in Fig. 2. Under this equivalence, the cross admittances $L_{lr}$ and $K_{lr}$ represent the thermoelectric ($Y = L$) and thermal conductance ($Y = K$) between the two contacts, coming from the *parallel* configuration of a resistor $R_{Y,lr}$ and a capacitor $C_{Y,lr}$ so that

$$Y_{lr}(\Omega) = \frac{1}{Z_{Y,lr}(\Omega)} = \frac{1}{R_{Y,lr}} + \frac{1}{1/i\Omega C_{Y,lr}}. \qquad (50)$$

On the other hand, the self admittances $L_\ell$ and $K_\ell$ represent the electrical/thermal conductances for charging/heating and relaxing in the contact $\ell$, coming from the *serial* configurations of a resistor $R_{Y,\ell}$ and a capacitor $C_{Y,\ell}$ so that

$$Y_\ell(\Omega) = \frac{1}{Z_{Y,\ell}(\Omega)} = \frac{1}{R_{Y,\ell} + 1/i\Omega C_{Y,\ell}}. \qquad (51)$$

Generally, the above resistances and capacitances are functions of the frequency $\Omega$. However, in the low-frequency ($\hbar\Omega \ll k_B T$) limit, they can be approximated to constants. We also assume the low-temperature ($k_B T \ll \Gamma_\ell$) condition to apply the Sommerfeld approximation. The self resistances and capacitances are then approximated to

$$C_{L,\ell} = -ehG_{\text{th}}T\frac{\Gamma_\ell}{\Gamma}\sum_\sigma \rho'_\sigma(0), \tag{52a}$$

$$R_{L,\ell} = \frac{1}{eG_{\text{th}}T}\frac{\Gamma}{\Gamma_\ell}\frac{\sum_\sigma \rho'_\sigma(0)\rho_\sigma(0)}{[\sum_\sigma \rho'_\sigma(0)]^2}, \tag{52b}$$

and

$$C_{K,\ell} = hG_{\text{th}}T\frac{\Gamma_\ell}{\Gamma}\sum_\sigma \rho_\sigma(0), \tag{53a}$$

$$R_{K,\ell} = -\frac{1}{2G_{\text{th}}T}\frac{\Gamma}{\Gamma_\ell}\frac{\sum_\sigma [\rho_\sigma(0)]^2}{[\sum_\sigma \rho_\sigma(0)]^2}, \tag{53b}$$

where $G_{\text{th}} \equiv \frac{\pi^2}{3}\frac{k_B^2 T}{h}$ is the thermal relaxation resistance for a single mode mesoscopic capacitor. On the other hand, the low-frequency cross resistances and capacitances are found to

$$R_{L,lr} = \frac{1}{(-e)hG_{\text{th}}T}\frac{\Gamma}{2\Gamma_l\Gamma_r}\frac{1}{\sum_\sigma \rho'_\sigma(0)}, \tag{54a}$$

$$C_{L,lr} = (-e)h^2 G_{\text{th}}T\frac{2\Gamma_l\Gamma_r}{\Gamma}\sum_\sigma \rho'_\sigma(0)\rho_\sigma(0), \tag{54b}$$

and

$$R_{K,lr} = \frac{1}{hG_{\text{th}}T}\frac{\Gamma}{2\Gamma_l\Gamma_r}\frac{1}{\sum_\sigma \rho_\sigma(0)}, \tag{55a}$$

$$C_{K,lr} = \frac{h^2 G_{\text{th}}T}{2}\frac{2\Gamma_l\Gamma_r}{\Gamma}\sum_\sigma [\rho_\sigma(0)]^2. \tag{55b}$$

The detailed analysis of the resistances and capacitances will be present in Sec. 4.2.

## 4.1 Why the choice of $\lambda = 1/2$ ?

In the original works [56, 57] which explored the heat transport in time domain, it is found that the meaningful definition of the contact energy should be determined to be $\mathcal{Q}_\ell(t)$ with $\lambda = 1/2$ [see Eq. (2)] because only this choice is consistent with the first and second laws of thermodynamics. While this argument alone justifies the choice of $\lambda = 1/2$, in this section we intend to find more evidence which supports the choice of $\lambda = 1/2$ by comparing our results for $I_\ell^c(\Omega)$ and $I_\ell^h(\Omega)$ with the previous ones obtained by different methods for the similar systems.

By using the equation-of-motion method, Rosselló, López, and Lim [60] have investigated the dynamical heat current through the similar setup as ours but with a single contact which is driven by an ac electric voltage. In calculating the heat current, they also took into account the energy barrier contribution with $\lambda = 1/2$, as proposed in Ref. [56]. As expected, our self thermoelectric admittance $L_\ell(\Omega)$ is exactly equal to their self electrothermal admittance $M_\ell(\Omega)/T$ [see Eq. (36) in Ref. [60]] which measures the heat current through the contact with respect to the ac electric driving in the contact. It justifies that the dynamical gravitational field in our setup should be coupled to $\mathcal{Q}_\ell$ with $\lambda = 1/2$ in order to get the correct dynamical charge current. Note that this agreement, $M_\ell(\Omega) = T L_\ell(\Omega)$ (where $T$ is the background temperature,

the common temperature in the contacts) also reflects the fact that the *reciprocal relation*, or the so-called Onsager's relation [61] should hold in the thermal transport. However, in the point of view of the fluctuation-dissipation theorem, both admittances $L_\ell(\Omega)$ and $M_\ell(\Omega)$ in the linear response regime are related to the same fluctuation

$$\langle [ \frac{d\mathcal{N}_\ell}{dt}, \frac{d\mathcal{Q}_\ell}{dt} ] \rangle . \tag{56}$$

In our setup, the perturbative (gravitational) field is coupled to $\mathcal{Q}_\ell$ and $\langle d\mathcal{N}_\ell/dt \rangle$ is measured, while in Rosselló's work the external (electric) field is coupled to $\mathcal{N}_\ell$ and $\langle d\mathcal{Q}_\ell/dt \rangle$ is measured. So, apparently, the agreement might be mathematically trivial because both used the same $\mathcal{Q}_\ell$ with $\lambda = 1/2$.

The second previous work to be compared with is the work done by Lim, López, and Sánchez [62] which has applied the *scattering theory* to the single-contact QD setup to obtain the dynamical charge and heat current in the linear response regime when the contact is driven either by an ac voltage or by an ac temperature. Note that in the scattering theory approach the barrier plays the role of the scatterer only, so no energy is stored in it and the contact energy is defined with respect to $\mathcal{H}_\ell$ only. They calculated the low-frequency responses of the currents with respect to the ac voltage and temperature, via the fluctuation-dissipation relation which they assumes to hold. We have found that our low-frequency expansion of the self admittances, Eqs. (52) and (53) are in good agreement with the scattering-theory predictions [see Eqs. (7), (8), and (9) in Ref. [62]]. This agreement strongly justifies our use of $\lambda = 1/2$, especially because they are derived from two different approaches: We have directly calculated the dissipative part by adopting the Luttinger's trick, while in Ref. [62] the admittances were obtained by calculating the fluctuations based on the scattering theory. Also, this agreement implies that the fluctuation-dissipation theorem holds for thermal transport, at least in the non-interacting and single-contact case.

## 4.2 More analysis on charge/heat currents

Since the physical discussion on the low-frequency self admittances, $R_{L/K,\ell}$ and $C_{L/K,\ell}$ has already been done in Ref. [62], we focus on the cross admittances here. First, the low-frequency and low-temperature expansions of the cross resistances $R_{L/K,lr}$ are physically reasonable and are in agreement with the scattering-theory prediction. For example, the low-temperature cross thermal conductance $1/R_{K,lr}$ [see the black dotted line in Fig. 3 (a)] was predicted to be proportional to the electric conductance $(e^2/h)(2\Gamma_l \Gamma_r / \Gamma) \sum_\sigma \rho_\sigma(0)$ [63], which is well reflected in Eq. (55), while the low-temperature cross thermoelectric conductance $1/R_{L,lr}$ [see the black dotted line in Fig. 3 (c)] is proportional to $\sum_\sigma \rho'_\sigma(0)$ as expected. It should be noted that the fluctuation-dissipation theorem applied to the heat transport through two-contact systems is no longer valid because scattering events that connect two different terminals induce a nonvanishing term for the equilibrium heat-heat correlation function at the low temperature limit, which is incompatible with the expected behavior of $K_{lr}(\Omega)$ [63, 64]. Therefore, one cannot exploit the fluctuation-dissipation theorem to study the dynamic heat transport through the quantum-dot systems described in terms of the tight-binding model. It signifies that our Luttinger formalism is the promising candidate for the systematic study of dynamical temperature driving.

Figure 3 displays the dot-level dependencies of the thermal/thermoelectric conductances and capacitances for a wide range of temperatures. The thermoelectric conductance $1/R_{L,lr}$ and capacitances $C_{L,lr}$ share similar dependence on the dot level, whose qualitative feature does not change much as the temperature increases [see Figs. 3 (c) and (d)]. While the thermal conductance $1/R_{K,lr}$ and capacitances $C_{K,lr}$ also share similar dependence on the dot level, they change from single-peak shape to double-peak one as the temperature increases [see

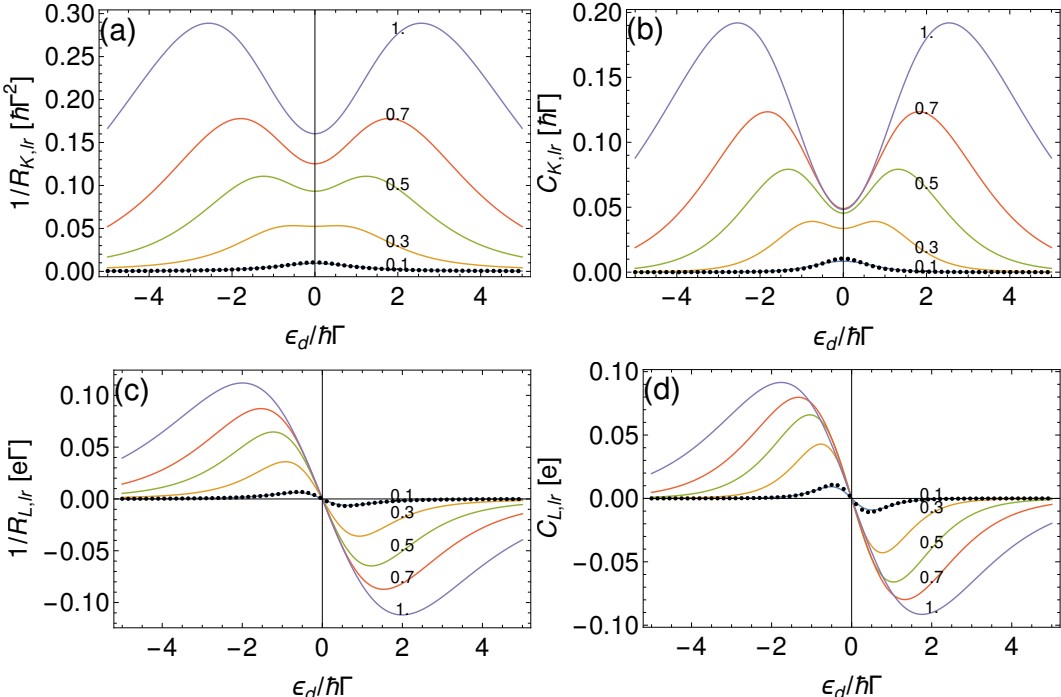

Figure 3: (a) The cross thermal conductance $1/R_{K,lr}$ and (b) the cross thermal capacitance $C_{K,lr}$, and (c) the cross thermoelectric conductance $1/R_{L,lr}$ and (d) the cross thermoelectric capacitance $C_{L,lr}$ in the low-frequency limit as functions of the spin-degenerate dot level $\epsilon_d$ in the noninteracting case with the symmetric coupling, $\Gamma_l = \Gamma_r$. Each curve corresponds to the different temperature $k_B T/\hbar\Gamma$ whose value is annotated. The resistances and capacitors for the cross admittances are defined via Eq. (50) and the corresponding cross thermal and thermoelectric admittances are evaluated via Eq. (48) by using given values of temperatures and the value of frequency chosen numerically as small as possible. The black dotted lines correspond to the low-temperature limits as given by Eqs. (54) and (55).

Figs. 3 (a) and (b)]. Recall that the heat current depends not only on the carrier occupation but also the carrier energy. At high temperatures, high-energy carrier can make more contribution to the heat current, which is the reason why the heat current can be larger at the off-resonant condition, as demonstrated in Figs. 3 (a) and (b).

Very interesting property peculiar to the noninteracting condition can be found in the RC times defined as

$$\tau_{Y,\ell} \equiv |R_{Y,\ell} C_{Y,\ell}| \quad \text{and} \quad \tau_{Y,lr} \equiv |R_{Y,lr} C_{Y,lr}|, \tag{57}$$

for $Y = L, K$. In the noninteracting case, the self and cross RC times are always equal to each other, that is,

$$\tau_{Y,l} = \tau_{Y,r} = \tau_{Y,lr}, \tag{58}$$

for both $Y = L, K$. It is because the self and cross admittances share the same frequency dependence: From Eqs. (47) and (48), one can find that $L_\ell(\Omega)$ and $L_{lr}(\Omega)$ are proportional to $\sum_\sigma P_{1\sigma}(\Omega)$, while $K_\ell(\Omega)$ and $K_{lr}(\Omega)$ are proportional to $\sum_\sigma P_{2\sigma}(\Omega)$. As we will see in the next section, this is not the case for interacting QD systems. That is, the comparison between the self and cross response times can be used to measure the effect of the interaction. Figure 4

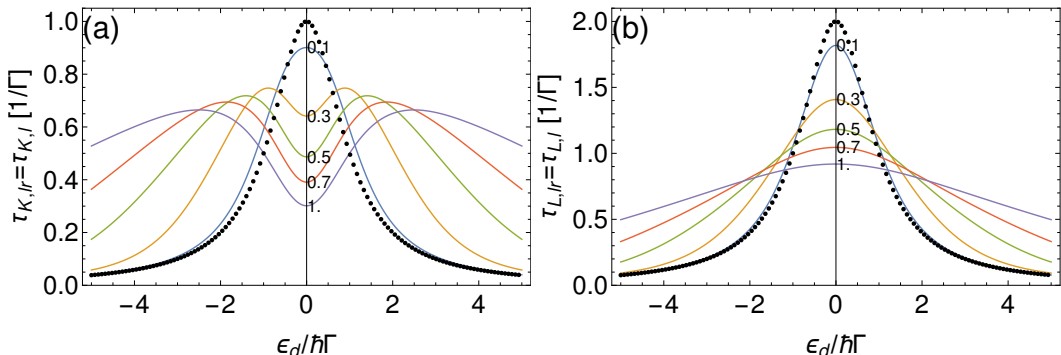

Figure 4: (a) The RC times $\tau_{K,lr} = \tau_{K,\ell}$ from thermal admittance and (b) the RC time $\tau_{L,lr} = \tau_{L,\ell}$ from thermoelectric admittance in the low-frequency limit as functions of the spin-degenerate dot level $\epsilon_d$ in the noninteracting case with the same parameters as in Fig. 3. The black dotted lines correspond to the low-temperature limits as given by Eq. (59).

shows the dot-level dependence of the RC times. The black dotted lines correspond to the low-temperature limit which is given by

$$\tau_{L,lr} = 2\tau_{K,lr} = h \frac{\sum_\sigma [\rho_\sigma(0)]^2}{\sum_\sigma \rho_\sigma(0)} . \tag{59}$$

While $\tau_{L,lr}$ shows the behavior similar to the QD density of states which broadens as the temperature increases, $\tau_{K,lr}$ features the double-peak structure at high temperatures.

Finally, the time-averaged power (38) for the noninteracting case is obtained as

$$\overline{P}(\Omega) = -\sum_\ell \frac{1}{2}\Psi_\ell^2 \Omega^2 C_{K,\ell}^2 R_{K,\ell} + \frac{1}{2}\frac{(\Psi_l - \Psi_r)^2}{R_{K,lr}} . \tag{60}$$

We then found that in the low-frequency limit only the second term remains finite so that only the cross thermal resistance $R_{K,lr}$ is responsible for the energy dissipation. Interestingly, this second term for the dissipation is identical to its electric counterpart, $V^2/2R$ where $V$ is the electric voltage drop and $R$ is the electric resistance between the two contacts. We again would like to stress that Eq. (60) is a natural outcome obtained by following the procedure based on our Luttinger formalism, without resorting to some heuristic arguments. Therefore our formalism is proven to provide a systematic way to investigate the dynamical heat transport in the linear response regime.

## 5 Interacting case: Hartree approximation

Our formalism is not limited to the noninteracting case. The charge and heat currents, Eqs. (41) and (37) can be calculated as long as the equilibrium and $n = 1$ components of the retarded/advanced QD Green's functions are provided. In this section we take into account the Coulomb interaction in the quantum dot which is now described by

$$\mathcal{H}_D = \sum_\sigma \epsilon_\sigma d_\sigma^\dagger d_\sigma + U n_\uparrow n_\downarrow. \tag{61}$$

Unfortunately, in the presence of finite Coulomb interaction, it is impossible to obtain any analytical form of the QD NEGFs without a proper approximation. Here we adopt the simplest

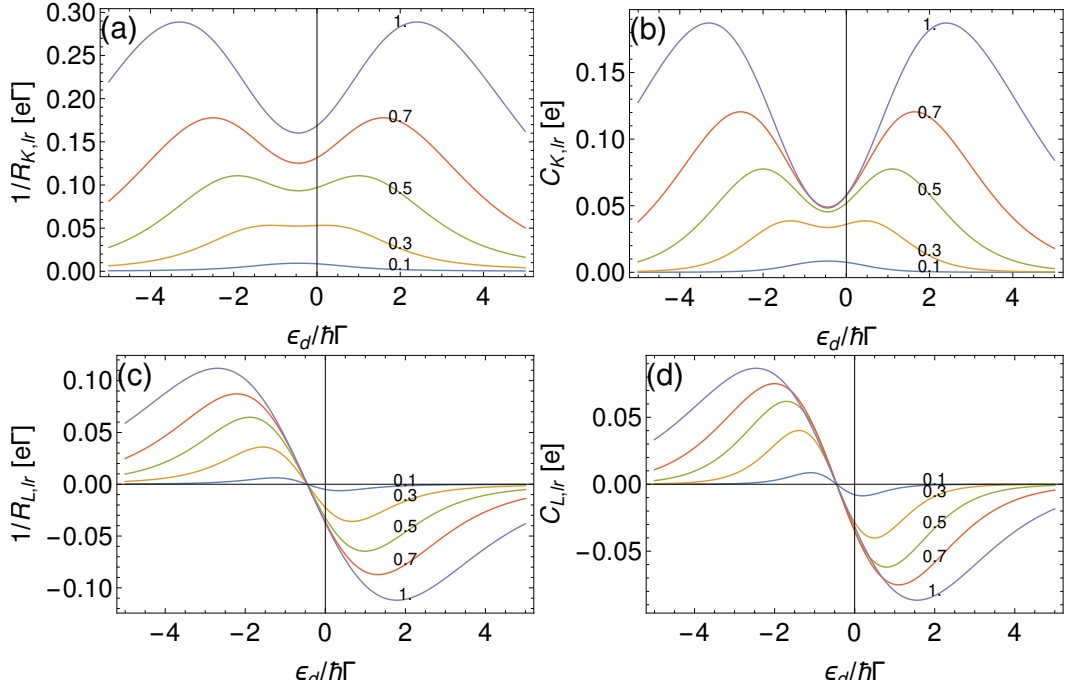

Figure 5: (a) The cross thermal conductance $1/R_{K,lr}$ and (b) the cross thermal capacitance $C_{K,lr}$, and (c) the cross thermoelectric conductance $1/R_{L,lr}$ and (d) the cross thermoelectric capacitance $C_{L,lr}$ in the low-frequency limit as functions of the spin-degenerate dot level $\epsilon_d$ in the interacting case with $U/\hbar\Gamma = 0.9$ and the symmetric coupling, $\Gamma_l = \Gamma_r$. Each curve corresponds to the different temperature $k_B T/\hbar\Gamma$ whose value is annotated.

approximation: the *Hartree approximation* which reduces the two-particle correlations to one-particle ones as

$$\langle \{d_\sigma(t), n_{\bar\sigma}(t')d_\sigma(t')\} \rangle \approx \langle n_{\bar\sigma}(t') \rangle \langle \{d_\sigma(t), d_\sigma(t')\} \rangle . \tag{62}$$

Then, the Dyson's equations for the QD NEGFs are found to be basically similar to those for the noninteracting case but with the retarded/advanced self energies being now replaced by

$$\Sigma_{\sigma,\mathrm{HF}}^{R/A}(t, t') = \Sigma_\sigma^{R/A}(t, t') + \delta(t - t')\frac{U}{\hbar} \langle n_{\bar\sigma}(t) \rangle . \tag{63}$$

Note that lesser self energy $\Sigma_\sigma^<(t, t')$ remains unchanged compared to the noninteracting case. This additional term in $\Sigma_{\sigma,\mathrm{HF}}^{R/A}(t, t')$ induces two changes compared to the noninteracting case: (1) The effective dot level is shifted from the unperturbed one,

$$\epsilon_\sigma \to \epsilon_{\sigma,\mathrm{HF}} = \epsilon_\sigma + \frac{U}{\hbar}n_{\bar\sigma}(0), \tag{64}$$

where

$$n_\sigma(0) = \int \frac{d\omega}{2\pi} f(\omega)\mathcal{G}_\sigma^R(0, \omega)(2\Gamma)\mathcal{G}_\sigma^A(0, \omega), \tag{65}$$

is the equilibrium QD occupation which should be determined in a self-consistent way. Note that the equilibrium QD Green's functions now depend on $\epsilon_{\sigma,\mathrm{HF}}$:

$$\mathcal{G}_\sigma^{R/A}(0, \omega) = \frac{1}{\omega - \epsilon_{\sigma,\mathrm{HF}}/\hbar \pm i\Gamma} . \tag{66}$$

(2) The $n = 1$ Fourier component of the QD Green's functions are now finite:

$$\mathcal{G}_\sigma^{R/A}(1, \omega) = \mathcal{G}_\sigma^{R/A}(0, \omega + \Omega)\frac{U}{\hbar}n_{\bar{\sigma}}(1, \Omega)\mathcal{G}_\sigma^{R/A}(0, \omega), \tag{67}$$

where

$$n_\sigma(1, \Omega) = \int \frac{d\omega}{2\pi i}\mathcal{G}_\sigma^<(1, \omega), \tag{68}$$

is the $n = 1$ Fourier component of the QD occupation. By using the charge conservation, one can obtain the explicit expression of $n_\sigma(1, \Omega)$ which is found to be

$$n_\sigma(1, \Omega) = -\sum_\ell \frac{\Psi_\ell}{2}2\Gamma_\ell X_\sigma(\Omega), \tag{69}$$

with

$$X_\sigma(\Omega) \equiv \frac{P_{1\sigma}(\Omega) + \frac{2\Gamma U}{\hbar}P_{0\sigma}(\Omega)P_{1\bar{\sigma}}(\Omega)}{1 - \left(\frac{2\Gamma U}{\hbar}\right)^2 P_{0\sigma}(\Omega)P_{0\bar{\sigma}}(\Omega)}. \tag{70}$$

Then, following the recipe proposed in our formalism [see Appendix E for further explanations], the charge and heat currents can be obtained [see Eqs. (121a) and (121b)] and the corresponding self/cross thermoelectric and thermal admittances are found to be

$$\frac{L_\ell(\Omega)}{2i\Gamma_\ell} = \Omega e \sum_\sigma \left[P_{1\sigma}(\Omega) + \frac{2\Gamma U}{\hbar}P_{0\sigma}(\Omega)X_{\bar{\sigma}}(\Omega)\right], \tag{71a}$$

$$\frac{K_\ell(\Omega)}{2i\Gamma_\ell} = \Omega(-\hbar) \sum_\sigma \left[P_{2\sigma}(\Omega) + \frac{2\Gamma U}{\hbar}P_{1\sigma}(\Omega)X_{\bar{\sigma}}(\Omega)\right], \tag{71b}$$

and

$$\frac{L_{lr}(\Omega)}{4\Gamma_l\Gamma_r} = e \sum_\sigma \left[P_{1\sigma}(\Omega) + \frac{i\Omega U}{\hbar}P_{0\sigma}(\Omega)X_{\bar{\sigma}}(\Omega)\right], \tag{72a}$$

$$\frac{K_{lr}(\Omega)}{4\Gamma_l\Gamma_r} = (-\hbar) \sum_\sigma \left[P_{2\sigma}(\Omega) + \frac{i\Omega U}{\hbar}P_{1\sigma}(\Omega)X_{\bar{\sigma}}(\Omega)\right]. \tag{72b}$$

These admittances clearly display the corrections due to the Coulomb interaction, as shown in Fig. 5: The resonance is shifted and the curves are slightly deformed compared to the noninteracting case, but no qualitative changes are observed. For examples, the low-frequency and low-temperature cross resistances $R_{L,lr}$ and $R_{K,lr}$ are found to be identical to those in the noninteracting case except $\epsilon_\sigma$ being replaced by $\epsilon_{\sigma,\mathrm{HF}}$. Therefore, it is not convenient to find a solid evidence on the effect of the Coulomb interaction from the dot-level dependence of the resistances and capacitances.

Instead, we focus on the RC times. In the presence of the Coulomb interaction, the self and cross RC times are not equal to each other any longer:

$$\tau_{L,\ell} \neq \tau_{L,lr} \quad \text{and} \quad \tau_{K,\ell} \neq \tau_{K,lr}, \tag{73}$$

as demonstrated in Figs. 6 (a) and (b). The Coulomb corrections in $Y_\ell$ and $Y_{lr}$ ($Y = L, K$) are different [compare Eq. (71) with Eq. (72)] and make their frequency dependence different from each other. As matter of fact, the difference between them is proportional to $U$ for weak Coulomb interaction. By performing the low-frequency and low-temperature expansions

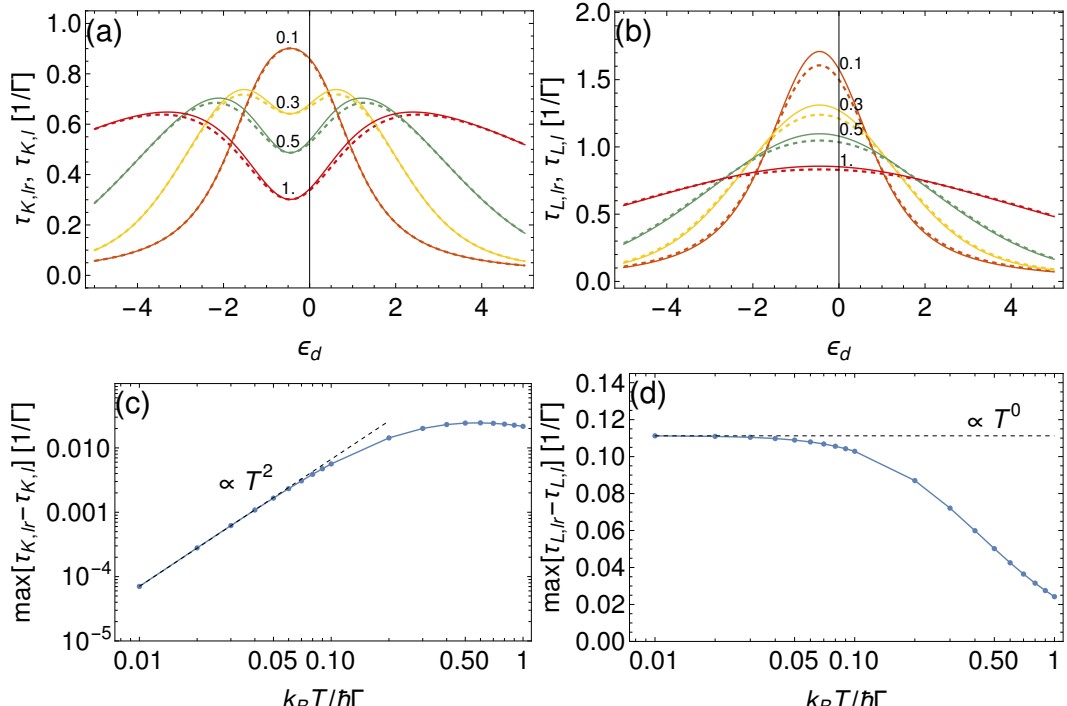

Figure 6: (a) The RC times $\tau_{K,lr}$ (solid lines) and $\tau_{K,l}$ (dotted lines) from thermal admittance and (b) the RC time $\tau_{L,lr}$ (solid lines) and $\tau_{L,l}$ (dotted lines) from thermoelectric admittance in the low-frequency limit as functions of the spin-degenerate dot level $\epsilon_d$ in the interacting case with the same parameters as in Fig. 5. (c) $\max_{\epsilon_d}[\tau_{K,lr} - \tau_{K,l}]$ and (d) $\max_{\epsilon_d}[\tau_{L,lr} - \tau_{L,l}]$ as functions of the temperature. The dotted lines are asymptotes in the $T \to 0$ limit.

similar to those done in the noninteracting case, one can find that, for spin-degenerate case with $\rho_\uparrow(0) = \rho_\downarrow(0) \equiv \rho_d$,

$$\tau_{L,lr} - \tau_{L,\ell} = \frac{hU}{2}\left(\frac{\rho_d^2}{1 - U\rho_d} - \frac{\rho_d}{h\Gamma}\right) + \mathcal{O}(T^2), \tag{74a}$$

$$\tau_{K,lr} - \tau_{K,\ell} = \frac{hU}{2}\frac{\pi^2}{3}(k_B T)^2 \rho_d'^2\left(1 - \frac{1}{h\Gamma\rho_d}\right) + \mathcal{O}(T^4). \tag{74b}$$

These expansions show that (1) both $\tau_{L,lr} - \tau_{L,\ell}$ and $\tau_{K,lr} - \tau_{K,\ell}$ are finite in the presence of the Coulomb interaction and (2) $\tau_{L,lr} - \tau_{L,\ell}$ saturates in the $T = 0$ limit, while $\tau_{K,lr} - \tau_{K,\ell}$ scales as $T^2$. These asymtotic behaviors for low temperatures are clearly manifested for a rather wide range of the temperature, as demonstrated in Figs. 6 (c) and (d). We expect that these temperature dependencies of the differences between the RC times can be used to identify the effect of the Coulomb interaction as long as $U$ is sufficiently small.

It should be noted that these temperature dependences due to the Coulomb corrections originate from the dynamical excitations reflected in the $n = 1$ components of the QD Green's functions, $\mathcal{G}_\sigma^{R/A<}(1, \omega)$, and not from the effective level shift in $\epsilon_{\sigma,\mathrm{HF}}$. Our Luttinger formalism takes into account the dynamical excitations systematically and correctly, even in the presence of the Coulomb interaction. Therefore, the utility of our formalism becomes more evident in the interacting systems.

In order to investigate the effect of the strong Coulomb interaction on the thermoelectric and thermal admittances, one should go beyond the Hartree approximation, taking into account the higher-order terms in the equation-of-motion method. For example, as long as the

Coulomb blockade is concerned, one can apply the Meir-Wingreen-Lee approximations [65,66] to our formalism, which will be our future work.

## 6 Conclusions

We have formulated a general formalism, based on the Luttinger's trick, to calculate the dynamic charge and heat currents through a multi-level quantum dot which is driven by time-dependent temperatures. Our Luttinger formalism is built on the correct definition of the dynamical contact energy and the requirement for the effective dot-contact coupling. It provides the general expressions for the linear-response charge and heat currents, given by Eqs. (34) and (37), respectively, which can be calculated as long as the $n = 0$ (equilibrium) and $n = 1$ Fourier components of the retarded/advanced and lesser QD Green's functions are known. Furthermore, with the help of charge conservation and the vanishing sum of the energy change rates, the knowledge of the lesser QD Green's functions can be avoided as long as some conditions are met.

The physically important point of our formalism is that it goes beyond the static and adiabatic condition for temperature modulation and it can capture naturally and systematically the effect due to dynamical excitations driven by the nonadiabatic temperature driving. Our formalism is then expected to be very adequate and useful, considering the state-of-the-art technology which enables very fast modulation of temperatures. Furthermore, our formalism works even in the presence of the Coulomb interaction and can reveal the role of the interaction in nonequilibrium thermal responses. Its application to the interacting case studied in Sec. 5 clearly demonstrates the success of our formalism in this regard, even though it has been done only in the Hartree approximation.

While the applications of our formalism present in this paper [see Secs. 4 and 5] are limited to the single-level quantum-dot system, it can be applied to any multi-level quantum-dot junctions so as to study the effect of multiple levels and spin-orbit interactions on the temperature-driven transport. Since our formalism allows the quantum dot to be any finite systems, it can also consider the case of multiple quantum dots. Furthermore, some interferometer-like geometry such as the junction with a quantum-embedded ring can be also incorporated into our formalism. As long as one ignores the Coulomb interactions in these systems, the equation-of-motion method can be readily exploited to derive the required retarded/advanced QD Green's functions and subsequently the dynamical charge and heat currents.

Real challenge is to take into account the Coulomb interaction in a non-perturbative way. There is no analytical solution in this case without proper approximations. Usually, it is very hard to obtain the exact form of equilibrium QD Green's functions let alone their dynamical ($n = 1$) Fourier components. One promising method to deal with the Coulomb interaction in a non-perturbative way is to use the numerical renormalization group [67] in order to obtain the QD Green's functions in a numerical way. While originally the numerical renormalization group is restricted to the equilibrium case, recently its extension, so-called the time-dependent numerical renormalization group, is improved to deal with the nonequilibrium case with a periodic driving [68,69], so that the two-time QD Green's functions are successfully calculated. While it has some issues about the accuracy, we believe that this method can be safely used to study the linear response regime in which the driving is sufficiently weak.

Another interesting theoretical challenge is to extend the Luttinger formalism beyond the linear regime. The original proposal of the Luttinger's trick [48] was based on the linear response regime so that the gradient of the gravitational field is identified with the gradient of the temperature. In fact, there is no physically reliable justification of the use of the Luttinger's trick beyond the linear regime. However, recently some tried to extend the Luttinger's idea

into the nonlinear-regime study of the nanostructures: the steady-state and transient behaviors of charge and heat currents after sudden quench of the gravitational field [50, 54] and temperature-driven adiabatic pumping [49, 55]. The predictions made by these works are quite interesting and nontrivial, while their validity of the prediction is uncertain and to be confirmed in experiments. So, knowing that there is no systematic way to deal with the time-dependent temperature, it may be very physically interesting to extend our formalism beyond the linear response regime so that a new physics is explored.

Finally, we would like to address briefly the experimental realization of our scheme. In order to test our predictions, a time-dependent modulation and control of temperature in the reservoirs should be experimentally implemented. We think, for example, that our theory may be tested in spin qubits [40] with detunnings of order of few meV that corresponds to ac frequencies for the temperature modulation of about hundreds of MHz. These are frequencies that are experimentally accessible nowadays. Alternatively, for higher frequencies (in the GHz and THz range), a design of time-dependent temperature signals by employing a collection of quantum harmonic oscillators that mediate the interactions between the quantum system and a thermal bath has been recently proposed in Ref. [47]. It is worth noting that in order to realize our prediction, the driving frequency should be not too low. It is because our scheme is based on the quantum-mechanically coherent state during the driving so the period of the driving should be shorter than the decoherence time, which in turn requires frequencies in the GHz regime in quantum-dot systems [19]. In short, the main hurdle to deal with is to maintain the quantum coherence long enough during the temperature modulation.

# Acknowledgments

**Funding information** R.L. acknowledges the financial support by the Grant No. PDR2020/12 sponsored by the Comunitat Autonoma de les Illes Balears through the Direcció General de Política Universitaria i Recerca with funds from the Tourist Stay Tax Law ITS 2017-006, the Grant No. PID2020-117347GB-I00, and the Grant No. LINKB20072 from the CSIC i-link program 2021. This work has been partially supported by the María de Maeztu project CEX2021-001164-M funded by the MCIN/AEI/10.13039/501100011033. P.S. acknowledges the financial support of the French National Research Agency (project SIM-CIRCUIT, ANR-18-CE47-0014-01). M.L. was supported by the National Research Foundation of Korea (NRF) grant funded by the Korea government (MSIT)(No.2018R1A5A6075964).

# A  Nonequilibrium Green functions and Dyson's equations

In our study we employ the nonequilibrium Keldysh formalism to express charge and heat current in terms of the QD Green's functions. It is convenient to recast the Green's functions in a matrix form as

$$\widehat{\mathcal{G}} = \begin{bmatrix} \mathcal{G}^t(t,t') & \mathcal{G}^<(t,t') \\ \mathcal{G}^>(t,t') & \mathcal{G}^{\bar{t}}(t,t') \end{bmatrix}, \tag{75}$$

where $\mathcal{G}^<_{\mathcal{A},\mathcal{B}}(t,t') = i\langle |\mathcal{B}^\dagger(t')\mathcal{A}(t)| \rangle$, $\mathcal{G}^<_{\mathcal{A},\mathcal{B}}(t,t') = -i\langle |\mathcal{A}(t)\mathcal{B}^\dagger(t')| \rangle$, and $\mathcal{G}^{t(\bar{t})}_{\mathcal{A},\mathcal{B}}(t,t') = -i\langle |\mathcal{T}_{c(\bar{c})}\mathcal{A}(t)\mathcal{B}^\dagger(t')| \rangle$ are the lesser, greater, and (anti-)time-ordered Green's functions between operators $\mathcal{A}$ and $\mathcal{B}$, respectively. The retarded and advanced Green's functions can be obtained through $\mathcal{G}^R = \mathcal{G}^t - \mathcal{G}^<$ and $\mathcal{G}^A = \mathcal{G}^< - \mathcal{G}^{\bar{t}}$. Here $\mathcal{A}$ and $\mathcal{B}$ can be either the dot operator $d_m$ or the contact operator $c_{\ell\mathbf{k}\sigma}$, and accordingly, the QD, contact, and QD-contact Green's functions are defined as given in Eqs. (17) and (18). We apply the equation-of-motion

technique with respect to the Hamiltonian (7). For convenience, we introduce the time-varying tunneling amplitudes (21) and contact energy (25) so that the contact and tunneling Hamiltonians can be written as

$$\mathcal{H}_{C\ell,\Psi}(t) = \sum_{\mathbf{k}\sigma} \epsilon_{\ell\mathbf{k}}(t) c^{\dagger}_{\ell\mathbf{k}\sigma} c_{\ell\mathbf{k}\sigma} \,, \tag{76a}$$

$$\mathcal{H}_{T\ell,\Psi}(t) = \sum_{m\mathbf{k}\sigma} \left( t_{\ell\mathbf{k}\sigma,m}(t) d^{\dagger}_m c_{\ell\mathbf{k}\sigma} + t^*_{\ell\mathbf{k}\sigma,m}(t) c^{\dagger}_{\ell\mathbf{k}\sigma} d_m \right) , \tag{76b}$$

apart from the time-dependent numbers which do not affect the dynamics of the Green's functions.

Via the equation-of-motion method, it is straightforward to obtain the following Dyson's equations:

$$\widehat{\mathcal{G}}_{m,\ell\mathbf{k}\sigma}(t,t') = \int dt'' \sum_{m''} \widehat{\mathcal{G}}_{m,m''}(t,t'') \frac{t_{\ell\mathbf{k}\sigma,m''}(t'')}{\hbar} \tau_3 \widehat{g}_{\ell\mathbf{k}\sigma}(t'',t') , \tag{77a}$$

$$\begin{aligned} \widehat{\mathcal{G}}_{\ell\mathbf{k}\sigma,\ell'\mathbf{k}'\sigma'}(t,t') = {} & \delta_{\ell\ell'}\delta_{\mathbf{k}\mathbf{k}'}\delta_{\sigma\sigma'} \widehat{g}_{\ell\mathbf{k}\sigma}(t,t') \\ & + \int dt'' \widehat{g}_{\ell\mathbf{k}\sigma}(t,t'')\tau_3 \sum_{m} \frac{t^*_{\ell\mathbf{k}\sigma,m}(t'')}{\hbar} \widehat{\mathcal{G}}_{m,\ell'\mathbf{k}'\sigma'}(t'',t') , \end{aligned} \tag{77b}$$

where $\tau_3$ is the third Pauli matrix in the Keldysh space and $\widehat{g}_{\ell\mathbf{k}\sigma}(t,t')$ denotes the *uncoupled* contact Green's function matrix as introduced in the main text. Explicitly, the uncoupled contact Green's functions are given by

$$g^{R/A}_{\ell\mathbf{k}\sigma}(t,t') = \mp i\Theta(\pm(t-t')) e^{-\frac{i}{\hbar}\int_{t'}^{t} dt'' \epsilon_{\ell\mathbf{k}}(t'')} , \tag{78a}$$

$$g^{<}_{\ell\mathbf{k}\sigma}(t,t') = e^{-\frac{i}{\hbar}\int_{t'}^{t} dt'' \epsilon_{\ell\mathbf{k}}(t'')} i f(\epsilon_{\ell\mathbf{k}}) , \tag{78b}$$

where $f(\epsilon)$ is the Fermi function at the temperature $T$ and the phase factor is evaluated as

$$e^{-\frac{i}{\hbar}\int_{t'}^{t} dt'' \epsilon_{\ell\mathbf{k}}(t'')} = e^{-\frac{i}{\hbar}\epsilon_{\ell\mathbf{k}}\left(t-t'+\widetilde{\Psi}_{\ell}(t)-\widetilde{\Psi}_{\ell}(t')\right)} , \tag{79}$$

with

$$\widetilde{\Psi}_{\ell}(t) = \int_{0}^{t} dt' \Psi_{\ell}(t') = \frac{\Psi_{\ell}}{\Omega} \sin \Omega t . \tag{80}$$

In order to evaluate the charge and heat currents, we need to get the expressions for $\mathcal{G}^{<}_{m,\ell\mathbf{k}\sigma}(t,t')$ and $\mathcal{G}^{<}_{\ell\mathbf{k}\sigma,\ell'\mathbf{k}'\sigma'}(t,t')$ [see Eqs. (19a), (20a), and (20b)] from the Dyson's equations (77a) and (77b):

$$\begin{aligned} \mathcal{G}^{<}_{m,\ell\mathbf{k}\sigma}(t,t') = \sum_{m''} \int dt'' \Big[ & \mathcal{G}^{R}_{mm''}(t,t'') \frac{t_{\ell\mathbf{k}\sigma,m''}(t'')}{\hbar} g^{<}_{\ell\mathbf{k}\sigma}(t'',t') \\ & + \mathcal{G}^{R}_{mm''}(t,t'') \frac{t_{\ell\mathbf{k}\sigma,m''}(t'')}{\hbar} g^{<}_{\ell\mathbf{k}\sigma}(t'',t') \Big] , \end{aligned} \tag{81a}$$

$$\begin{aligned} \mathcal{G}^{<}_{\ell\mathbf{k}\sigma,\ell\mathbf{k}\sigma}(t,t') = {} & g^{<}_{\ell\mathbf{k}\sigma}(t,t') \\ & + \sum_{m''m'''} \int dt'' \int dt''' \frac{t^*_{\ell\mathbf{k}\sigma,m''}(t'')}{\hbar} \Big( g^{R}_{\ell\mathbf{k}\sigma}(t,t'') \mathcal{G}^{R}_{m''m'''}(t'',t''') g^{<}_{\ell\mathbf{k}\sigma}(t''',t') \\ & + g^{<}_{\ell\mathbf{k}\sigma}(t,t'') \mathcal{G}^{A}_{m''m'''}(t'',t''') g^{A}_{\ell\mathbf{k}\sigma}(t''',t') \\ & + g^{R}_{\ell\mathbf{k}\sigma}(t,t'') \mathcal{G}^{<}_{m''m'''}(t'',t''') g^{A}_{\ell\mathbf{k}\sigma}(t''',t') \Big) \frac{t_{\ell\mathbf{k}\sigma,m'''}(t''')}{\hbar} . \end{aligned} \tag{81b}$$

We insert the above expressions into Eqs. (19a), (20a), and (20b) and obtain Eqs. (23) and (26) in terms of the relevant self energies $\Sigma_\ell^a$ defined as Eq. (24) and the self-energy-like terms $\Xi_\ell^{ab}$ defined as Eq. (27).

## B  Explicit expressions and linear expansions of $\Sigma_\ell^{R/A/<}(t,t')$ and $\Xi_\ell^{AR/<R/A<}(t,t',t'')$

To evaluate the charge and heat currents via Eqs. (23) and (26), one needs to know the explicit expressions for the self energies $\Sigma_\ell^a(t,t')$ [see Eq. (24)] and the self-energy-like forms $\Xi_\ell^{ab}(t,t',t'')$ [see Eq. (27)]. For simplicity, we take the wide-band limit with a constant density of states $\rho_0$ for both the contacts, which allows us to replace the sum $\sum_{\mathbf{k}} F(\epsilon_{\ell\mathbf{k}})$ by the integral $\rho_0 \int_{-\infty}^{\infty} d\epsilon\, F(\epsilon)$.

Using the explicit expressions for $g_{\ell\mathbf{k}\sigma}^{R/A/<}(t,t')$, one can express the self energies $\Sigma_\ell^{R/A/<}(t,t')$ as

$$\Sigma_\ell^{R/A}(t,t') = \mp 2i\mathbf{\Gamma}_\ell \Theta(\pm(t-t'))(1+\lambda\Psi_\ell(t))(1+\lambda\Psi_\ell(t')) \int \frac{d\omega}{2\pi} e^{-i\omega(t-t'+\widetilde{\Psi}_\ell(t)-\widetilde{\Psi}_\ell(t'))},  \quad (82a)$$

$$\Sigma_\ell^{<}(t,t') = 2i\mathbf{\Gamma}_\ell (1+\lambda\Psi_\ell(t))(1+\lambda\Psi_\ell(t')) \int \frac{d\omega}{2\pi} f(\omega) e^{-i\omega(t-t'+\widetilde{\Psi}_\ell(t)-\widetilde{\Psi}_\ell(t'))},  \quad (82b)$$

where $\omega = \epsilon/\hbar$ and we have used $f(\omega)$ instead of $f(\hbar\omega)$, for simplicity. The integration over $\omega$ can be done for $\Sigma_{\ell\sigma}^{R/A}$, giving rise to

$$\Sigma_\ell^{R/A}(t,t') = \mp i\mathbf{\Gamma}_\ell \frac{(1+\lambda\Psi_\ell(t))^2}{1+\Psi_\ell(t)} \delta(t-t'),  \quad (83)$$

where we have used the fact that $|\Psi_\ell| \leq 1$: The temperature oscillation amplitude cannot be larger than the base temperature $T$ itself. The corresponding Fourier components, $\Sigma_{\ell,n}^{R/A}(\omega)$ can be also obtained in an analytical form (we do not write down the detailed expression here) which involves the regularized generalized hypergeometric functions. However, unfortunately, no simple analytical expressions for $\Sigma_\ell^{<}(t,t')$ nor $\Sigma_{\ell,n}^{<}(\omega)$ are available. Specifically, by using the identity $e^{i\beta \sin\Omega t} = \sum_{n=-\infty}^{\infty} e^{i\Omega t} J_n(\beta)$ where $J_n(x)$ are the first kind Bessel functions and with a help of recurrence relations for $J_n(x)$, one can obtain

$$\Sigma_{\ell,n}^{<}(\omega) = 2i\mathbf{\Gamma}_\ell \sum_{m=-\infty}^{\infty} f(\omega_m) J_{n+m}(\frac{\Psi_\ell}{\Omega}\omega_m) J_m(\frac{\Psi_\ell}{\Omega}\omega_m)\left(1+\frac{\lambda(n+m)}{\omega_m}\Omega\right)\left(1+\frac{\lambda m}{\omega_m}\Omega\right),  \quad (84)$$

with $\omega_m \equiv \omega - m\Omega$. For general value of $\Psi_\ell$, this expression is not adequate for analytical nor numerical analysis since it requires the summation over $m$ from $-\infty$ to $\infty$.

The situation becomes worse for $\Xi_\ell^{ab}(t,t',t'')$ whose explicit expressions involve more complicated integration:

$$\Xi_\ell^{AR}(t,t',t'') = -2i\hbar\Gamma_\ell\Theta(t-t')\Theta(t-t'')(1+\lambda\Psi_\ell(t'))(1+\lambda\Psi_\ell(t''))$$
$$\times \int \frac{d\omega}{2\pi} \omega e^{-i\omega(t'-t''+\widetilde{\Psi}_\ell(t')-\widetilde{\Psi}_\ell(t''))}, \tag{85a}$$

$$\Xi_\ell^{<R}(t,t',t'') = -2i\hbar\Gamma_\ell\Theta(t-t'')(1+\lambda\Psi_\ell(t'))(1+\lambda\Psi_\ell(t''))$$
$$\times \int \frac{d\omega}{2\pi} \omega f(\omega) e^{-i\omega(t'-t''+\widetilde{\Psi}_\ell(t')-\widetilde{\Psi}_\ell(t''))}, \tag{85b}$$

$$\Xi_\ell^{A<}(t,t',t'') = \left[\Xi_\ell^{<R}(t,t'',t')\right]^\dagger. \tag{85c}$$

Unfortunately, for general values of $\Psi_\ell$, the integrations over $\omega$ and the Fourier transformation do not yield any manageable analytical expressions. On the other hand, we have found that the linear expansion of the self energies with respect to $\Psi_\ell$ yields reliable and analytical expressions. Therefore, in our study we focus on the linear response regime.

Before presenting the linear expansion of the self energies, the reliability of the linear expansion should be examined. The linear expansion in $\Psi_\ell$ approximates the exponential function in the integrals, Eqs. (82) and (85) as

$$e^{-i\omega(\widetilde{\Psi}_\ell(t)-\widetilde{\Psi}_\ell(t'))} \approx 1 - i\omega(\widetilde{\Psi}_\ell(t)-\widetilde{\Psi}_\ell(t')), \tag{86}$$

before the integration over $\omega$ is done. While $\Psi_\ell$ is assumed to be small enough, in fact, $\omega\Psi_\ell$ is not small for large values of $\omega$ which definitely happen during the integration over $\omega$, which may disqualify the use of the linear expansion. However, we have confirmed that this expansion produces correct results. Our justifications are two-fold. First, in the linear response regime, only the contact excitations close to the Fermi level are relevant: Note that $\epsilon = \hbar\omega$ is the contact excitation energy. Hence, the correctness of the approximation at higher energies does not matter. Secondly, we have explicitly adopted a *regularization function* $\eta(\omega)$ to the gravitational field so that the thermal driving is really effective only to the low-energy states: $\Psi_\ell \to \eta(\omega)\Psi_\ell$. Specifically, we have chosen a Gaussian regularization $\eta(\omega) = \exp[-(\omega/\omega_0)^2]$ with a constant $\omega_0$ which determines the range of energies to be meaningfully coupled to the gravitational field. Then, the linear expansion is well justified because $\omega\eta(\omega)\widetilde{\Psi}(t)$ is small for all values of $\omega$: Note that $\eta(\omega)$ decreases exponentially with $\omega$. Then, at the final stage of the calculations, we take $\omega_0 \to \infty$ to restore the original coupling of the gravitational field. We have confirmed that the result obtained from the regularization and taking the limit of $\omega_0 \to \infty$ is identical to that obtained by using the linear expansion, Eq. (86) from the beginning.

Applying the linear expansion (86) and performing the Fourier transformation (28) to $\Sigma_\ell^{R/A/<}(t,t')$, one can obtain

$$\Sigma_\ell^{R/A}(0,\omega) = \mp i\Gamma_\ell,$$
$$\Sigma_\ell^{R/A}(1,\omega) = \Sigma_{\ell\sigma}^{R/A}(-1,\omega) = \mp\left(\lambda-\frac{1}{2}\right)(2i\Gamma_\ell)\frac{\Psi_\ell}{2}, \tag{87}$$

and

$$\Sigma_\ell^{<}(0,\omega) = f(\omega)(2i\Gamma_\ell),$$
$$\Sigma_\ell^{<}(\pm1,\omega) = -\Delta_f(\omega\pm\Omega,\omega)\left(\omega\pm\frac{\Omega}{2}\right)(2i\Gamma_\ell)\frac{\Psi_\ell}{2}$$
$$+\left(\lambda-\frac{1}{2}\right)(f(\omega\pm\Omega)+f(\omega))(2i\Gamma_\ell)\frac{\Psi_\ell}{2}, \tag{88}$$

where the definition of $\Delta_f(\omega, \omega')$ is given by Eq. (36). As one can see, in the linear response regime, the $n = 0$ Fourier components are nothing but the equilibrium values at $\Psi_\ell = 0$, and the $n = \pm 1$ components are linear in $\Psi_\ell$ while all the higher components ($|n| \geq 2$) vanish up to the linear order in $\Psi_\ell$. One can note that the $n = \pm 1$ components become a lot simpler at $\lambda = 1/2$: In particular, $\Sigma_\ell^{R/A}(\pm 1, \omega)$ vanish at $\lambda = 1/2$.

The linear expansion is now applied to $\Xi_\ell^{ab}(n, n', \omega)$ [see Eq. (30)]. Up to the linear order in $\Psi_\ell$, only the Fourier components with $|n| \leq 1$, $|n'| \leq 1$ and $|n + n'| \leq 1$ are relevant:

$$
\begin{aligned}
\Xi_\ell^{AR}(0, 0, \omega) &= -I_\infty \hbar\omega(2i\Gamma_\ell), \\
\Xi_\ell^{AR}(\pm 1, 0, \omega) &= \pm 2(1 - \lambda)\frac{i}{\Omega}\hbar\omega(2i\Gamma_\ell)\frac{\Psi_\ell}{2}, \\
\Xi_\ell^{AR}(0, \pm 1, \omega) &= \mp\frac{i}{\Omega}\hbar\left(\omega \pm \frac{\Omega}{2}\right)(2i\Gamma_\ell),
\end{aligned}
\tag{89}
$$

and

$$
\begin{aligned}
\Xi_\ell^{<R}(0, 0, \omega) &= -I_\infty f(\omega)\hbar\omega(2i\Gamma_\ell), \\
\Xi_\ell^{<R}(\pm 1, 0, \omega) &= \pm\frac{i}{\Omega}\Delta_{\omega f}(\omega \mp \Omega, \omega)\hbar\left(\omega \mp \frac{\Omega}{2}\right)(2i\Gamma_\ell)\frac{\Psi_\ell}{2} \\
&\quad \mp\left(\lambda - \frac{1}{2}\right)\frac{i}{\Omega}[f(\omega)\hbar\omega + f(\omega \mp \Omega)\hbar(\omega \mp \Omega)](2i\Gamma_\ell)\frac{\Psi_\ell}{2}, \\
\Xi_\ell^{<R}(0, \pm 1, \omega) &= \mp\frac{i}{\Omega}f(\omega)\hbar\omega(2i\Gamma_\ell),
\end{aligned}
\tag{90}
$$

and

$$
\begin{aligned}
\Xi_\ell^{A<}(0, 0, \omega) &= [\Xi_\ell^{<R}(0, 0, \omega)]^\dagger, \\
\Xi_\ell^{A<}(\pm 1, 0, \omega) &= [\Xi_\ell^{<R}(\mp 1, 0, \omega)]^\dagger, \\
\Xi_\ell^{A<}(0, \pm 1, \omega) &= \left[\Xi_\ell^{<R}(0, \mp 1, \omega \pm \Omega)\right]^\dagger,
\end{aligned}
\tag{91}
$$

with $I_\infty \equiv \int_{-\infty}^{\infty} dt\, \Theta(-t)$. Note that owing to the constant $I_\infty$ the equilibrium contributions, $\Xi_\ell^{RA/R</<A}(0, 0, \omega)$ are divergingly large. However, we have found that these diverging contributions cancel out each other exactly in the zeroth-order term of the linear expansion of Eq. (26):

$$
\begin{aligned}
(\langle\mathcal{H}_{C\ell}\rangle - E_{C\ell 0})^{(0)} &= \int \frac{d\omega}{2\pi} \mathrm{Tr}\Bigg[\Xi_\ell^{AR}(0, 0, \omega)\mathbf{G}^<(0, \omega) \\
&\qquad + \Xi_\ell^{<R}(0, 0, \omega)\mathbf{G}^R(0, \omega) + \Xi_\ell^{A<}(0, 0, \omega)\mathbf{G}^A(0, \omega)\Bigg] \\
&= I_\infty \int \frac{d\omega}{2\pi}\hbar\omega\, \mathrm{Tr}\Big[(2i\Gamma_\ell)\big(\mathbf{G}^<(0, \omega) + f(\omega)\big(\mathbf{G}^R(0, \omega) - \mathbf{G}^A(0, \omega)\big)\big)\Big] \\
&= 0,
\end{aligned}
$$

where in the last line we have used the equality,

$$
\mathbf{G}^<(0, \omega) + f(\omega)\big(\mathbf{G}^R(0, \omega) - \mathbf{G}^A(0, \omega)\big) = 0,
\tag{92}
$$

which holds generally *in equilibrium* for any *interacting* quantum-dot junctions.

## C  Linear regime: Charge and heat currents and sum rules

The charge currents in the linear regime can be obtained by expressing (23a) in terms of the Fourier components $\mathbf{G}(n,\omega)$ and $\Sigma_\ell(n,\omega)$ via (29a) and then by keeping terms up to the linear order in $\Psi_\ell$. It is then quite straightforward to obtain the explicit expression, giving rise to Eq. (34). In the derivation, we have used the expressions for $\Sigma_\ell^{R/A/<}(n,\omega)$ [see Eqs. (87) and (88)] with $\lambda = 1/2$.

Our system conserves the total charge so that the charge conservation (40) holds. As long as the condition (39) holds, the charge conservation condition can be used to remove $\int d\omega \, \mathrm{Tr}[\mathbf{G}^<(1,\omega)]$ from the expression for the charge current. By inserting the expressions of the QD charge current (35) and the contact charge current (34) into the charge conservation (40), one can solve the integral $\int d\omega \, \mathrm{Tr}[\mathbf{G}^<(1,\omega)]$ in terms of other QD Green's functions: with $\lambda = 1/2$,

$$
\int \frac{d\omega}{2\pi} \mathrm{Tr}[\mathbf{G}^<(1,\omega)]
$$
$$
= -\int \frac{d\omega}{2\pi} \frac{1}{2i\Gamma + \Omega} \Bigg[ -\sum_\ell (2i\Gamma_\ell) \frac{\Psi_\ell}{2} \Delta_f(\omega+\Omega,\omega) \left(\omega + \frac{\Omega}{2}\right) \mathrm{Tr}\Big[\mathbf{G}^R(0,\omega+\Omega) - \mathbf{G}^A(0,\omega)\Big]
$$
$$
+ f(\omega+\Omega)(2i\Gamma) \mathrm{Tr}\Big[\mathbf{G}^R(1,\omega+\Omega) - \mathbf{G}^A(1,\omega)\Big] \Bigg].
\tag{93}
$$

By inserting Eq. (93) into the contact charge current (34), one gets Eq. (41).

For obtaining the heat currents we first take the average energies that are then expanded into

$$
\langle \mathcal{H}_{\mathrm{T}\ell} \rangle = E_{\mathrm{T}\ell 0}(1 - \lambda \Psi_\ell(t)) + E_{\mathrm{T}\ell}(\Omega)e^{-i\Omega t} + E_{\mathrm{T}\ell}(-\Omega)e^{i\Omega t}, \tag{94a}
$$

$$
\langle \mathcal{H}_{\mathrm{C}\ell} \rangle = E_{\mathrm{C}\ell 0} + E_{\mathrm{C}\ell}(\Omega)e^{-i\Omega t} + E_{\mathrm{C}\ell}(-\Omega)e^{i\Omega t}, \tag{94b}
$$

with

$$
E_{\mathrm{T}\ell}(\Omega) = \hbar \int \frac{d\omega}{2\pi i} \mathrm{Tr}\Bigg[ \Sigma_\ell^<(1,\omega)\big(\mathbf{G}^R(0,\omega+\Omega) + \mathbf{G}^A(0,\omega)\big)
$$
$$
+ f(\omega+\Omega)(2i\boldsymbol{\Gamma}_\ell)\big(\mathbf{G}^R(1,\omega+\Omega) + \mathbf{G}^A(1,\omega)\big) \Bigg], \tag{95}
$$

and

$$
E_{\mathrm{C}\ell}(\Omega) = \int \frac{d\omega}{2\pi} \mathrm{Tr}\Bigg[ \Xi_\ell^{<R}(1,0,\omega+\Omega)\big(\mathbf{G}^R(0,\omega+\Omega) - \mathbf{G}^A(0,\omega)\big) + \Xi_\ell^{AR}(1,0,\omega)\mathbf{G}^<(0,\omega)
$$
$$
+ \Xi_\ell^{<R}(0,1,\omega+\Omega)\big(\mathbf{G}^R(1,\omega+\Omega) - \mathbf{G}^A(1,\omega)\big) + \Xi_\ell^{AR}(0,1,\omega)\mathbf{G}^<(1,\omega) \Bigg], \tag{96}
$$

and the tunneling barrier energy at equilibrium

$$
E_{\mathrm{T}\ell 0} = \hbar \int \frac{d\omega}{2\pi i} f(\omega) \mathrm{Tr}\Big[(2i\boldsymbol{\Gamma}_\ell)\big(\mathbf{G}^R(0,\omega) + \mathbf{G}^A(0,\omega)\big)\Big]. \tag{97}
$$

Then, from Eqs. (13) and (22), the contact heat currents and the energy change rates in the linear regime are expressed in terms of $E_{\mathrm{C}/\mathrm{T}\ell}(\Omega)$:

$$
W_{\mathrm{C}\ell}(\Omega) = -i\Omega E_{\mathrm{C}\ell}(\Omega), \tag{98a}
$$

$$
W_{\mathrm{T}\ell}(\Omega) = -i\Omega \left( E_{\mathrm{T}\ell}(\Omega) - \frac{\Psi_\ell}{2}\lambda E_{\mathrm{T}\ell 0} \right), \tag{98b}
$$

$$
I_\ell^h(\Omega) = W_{\mathrm{C}\ell}(\Omega) + \lambda W_{\mathrm{T}\ell}(\Omega), \tag{98c}
$$

where the Fourier components of the energy change rates in the linear regime are defined as

$$W_{\mathrm{C/T}\ell}(t) = W_{\mathrm{C/T}\ell}(\Omega)e^{-i\Omega t} + W_{\mathrm{C/T}\ell}(-\Omega)e^{i\Omega t} \,. \tag{99}$$

Employing these expressions and for $\lambda = 1/2$ the Fourier component of the heat current in the linear regime is obtained as (37).

We have another similar sum rule for the energy change rates, Eq. (16), which is written in the linear regime as Eq. (43). While $W_{\mathrm{D}}(t) = d\langle\mathcal{H}_{\mathrm{D}}\rangle/dt = (i/\hbar)\langle[\mathcal{H},\mathcal{H}_{\mathrm{D}}]\rangle$ requires the specification of $\mathcal{H}_{\mathrm{D}}$, the other terms in the sum rule can be written as

$$
\begin{aligned}
\sum_{\ell}(W_{\mathrm{C}\ell}(\Omega) + W_{\mathrm{T}\ell}(\Omega)) &= \sum_{\ell}(-i\Omega)\left(E_{\mathrm{C}\ell}(\Omega) + E_{\mathrm{T}\ell}(\Omega) - \frac{1}{2}E_{\mathrm{T}\ell 0}\frac{\Psi_\ell}{2}\right) \\
&= \sum_{\ell}\hbar\int\frac{d\omega}{2\pi}\,\mathrm{Tr}\bigg[(2i\mathbf{\Gamma}_\ell)\bigg(\frac{\Psi_\ell}{2}\Delta_f(\omega+\Omega,\omega)\left(\omega+\frac{\Omega}{2}\right)^2\big(\mathbf{G}^R(0,\omega+\Omega) - \mathbf{G}^A(0,\omega)\big) \\
&\qquad + \frac{\Psi_\ell}{2}\frac{\Omega}{2}\Delta_f(\omega+\Omega,\omega)\left(\omega+\frac{\Omega}{2}\right)\big(\mathbf{G}^R(0,\omega+\Omega) + \mathbf{G}^A(0,\omega)\big) \\
&\qquad - (\omega+\Omega)f(\omega)\mathbf{G}^R(1,\omega) + \omega f(\omega+\Omega)\mathbf{G}^A(1,\omega) - \left(\omega+\frac{\Omega}{2}\right)\mathbf{G}^<(1,\omega)\bigg)\bigg].
\end{aligned}
\tag{100}
$$

This expression can be used to write the integral $\int d\omega\,\omega\,\mathrm{Tr}[\mathbf{G}^<(1,\omega)]$ in terms of other QD Green's functions, via the sum rule (43).

Finally, we find the expression for the power of dissipation (14) in the linear response regime. Up to the lowest order in $\Psi_\ell$, the power reads

$$P(t) = \sum_{\ell}\dot{\Psi}_\ell(t)\left[e^{-i\Omega t}\left(E_{\mathrm{C}\ell}(\Omega) + \frac{E_{\mathrm{T}\ell}(\Omega)}{2}\right) + (c.c.)\right], \tag{101}$$

which is of the second order in $\Psi_\ell$.

# D  Noninteracting Case: QD Green Functions and Charge/Heat Currents

By applying the equation-of-motion method to the noninteracting Hamiltonian (44), one can obtain the Dyson's equation for the QD Green's functions

$$\widehat{\mathcal{G}}_\sigma(t,t') = \widehat{g}_\sigma(t,t') + \sum_{\ell\mathbf{k}}\int dt''\widehat{\mathcal{G}}_{\sigma,\ell\mathbf{k}\sigma}(t,t'')\frac{t^*_{\ell\mathbf{k}\sigma,\sigma}(t'')}{\hbar}\tau_3\widehat{g}_\sigma(t'',t'), \tag{102}$$

where $\widehat{g}_\sigma(t,t')$ are the unpertubed QD Green's functions whose explicit expressions are identical to those of the uncoupled contact Green's functions (78) with the replacement of $\epsilon_{\ell\mathbf{k}}(t)$ by $\epsilon_\sigma$. By combining Eqs. (102) and (77a), one can find the Dyson's equation for the QD Green's functions in terms of the self energies (82),

$$\widehat{\mathcal{G}}_\sigma(t,t') = \widehat{g}_\sigma(t,t') + \int dt''\int dt'''\widehat{\mathcal{G}}_\sigma(t,t'')\tau_3\widehat{\Sigma}_\sigma(t'',t''')\tau_3\widehat{g}_\sigma(t''',t'), \tag{103}$$

or, more specifically, the equations for retarded/advanced QD Green's functions,

$$\mathcal{G}^{R/A}_\sigma(t,t') = g^{R/A}_\sigma(t,t') + \int dt''\int dt'''\mathcal{G}^{R/A}_\sigma(t,t'')\Sigma^{R/A}_\sigma(t'',t''')g^{R/A}_\sigma(t''',t'), \tag{104}$$

For the linear expansion, it is convenient to express the Dyson's equation in frequency domain so that

$$\mathcal{G}_\sigma^{R/A}(t,\omega) = \left(1 + \sum_n e^{in\Omega t} \mathcal{G}_\sigma^{R/A}(t,\omega + n\Omega)\Sigma_\sigma^{R/A}(n,\omega)\right) g_\sigma^{R/A}(\omega). \tag{105}$$

By using the linear expansions (87) of $\Sigma_\sigma^{R/A}(n,\omega)$ and by keeping up to the linear order in $\Psi_\ell$, the $n = 0$ (equilibrium) and $n = 1$ components of the retarded/advanced QD Green's functions are found to be

$$\mathcal{G}_\sigma^{R/A}(0,\omega) = \frac{1}{[g_\sigma^{R/A}(\omega)]^{-1} \pm i\Gamma}, \tag{106a}$$

$$\mathcal{G}_\sigma^{R/A}(1,\omega) = \mathcal{G}_\sigma^{R/A}(0,\omega + \Omega)\Sigma_\sigma^{R/A}(1,\omega)\mathcal{G}_\sigma^{R/A}(0,\omega), \tag{106b}$$

with $\Sigma_\sigma^{R/A}(1,\omega) \equiv \sum_\ell \Sigma_{\ell,\sigma}^{R/A}(1,\omega)$. Note that at $\lambda = 1/2$, $\Sigma_{\ell,\sigma}^{R/A}(1,\omega)$ is zero up to the linear order in $\Psi_\ell$ so that $\mathcal{G}_\sigma^{R/A}(1,\omega)$ also vanishes. Then, by exploiting the properties of the equilibrium noninteracting QD Green's functions (45) and the vanishingness of $\Sigma_{\ell,\sigma}^{R/A}(1,\omega)$, one can get the explicit expression for the charge current from the general formula of the charge current (41):

$$I_\ell^c(\Omega) = \Upsilon_\ell(-e)\sum_\sigma P_{1\sigma}(\Omega), \tag{107}$$

with

$$\Upsilon_\ell \equiv -\frac{\Psi_\ell}{2}(2i\Gamma_\ell)\Omega + \frac{\Psi_\ell - \Psi_{\bar{\ell}}}{2}(4\Gamma_l\Gamma_r), \tag{108}$$

with $\Psi_{\bar{l}} = \Psi_r$ and vice versa.

In order to find the explicit expression for the heat current, one should know the integrals of $\mathcal{G}_\sigma^<(1,\omega)$, that is, $\int d\omega\, \mathcal{G}_\sigma^<(1,\omega)$ and $\int d\omega\, \omega \mathcal{G}_\sigma^<(1,\omega)$. In the noninteracting case, one can easily derive the linear expansions of the lesser QD Green's functions $\mathcal{G}_\sigma^<(t,\omega)$ from the Dyson's equation (103). However, as proposed in the main text, we instead exploit the charge conservation (93) and the sum rule (43) for the energy change rates in order to express the integrals of $\mathcal{G}_\sigma^<(1,\omega)$, that is, $\int d\omega\, \mathcal{G}_\sigma^<(1,\omega)$ and $\int d\omega\, \omega \mathcal{G}_\sigma^<(1,\omega)$ in terms of the equilibrium retarded/advanced QD Green's functions. Below we derive their explicit expressions. From the charge conservation (93), by using the properties of the equilibrium QD Green's functions (45) and $\mathcal{G}_\sigma^{R/A}(1,\omega) = 0$, one can get

$$\int \frac{d\omega}{2\pi} \mathcal{G}_\sigma^<(1,\omega) = -\sum_\ell \frac{\Psi_\ell}{2}(2i\Gamma_\ell) P_{1\sigma}(\Omega). \tag{109}$$

Refer the definition of $P_{1\sigma}(\Omega)$ to Eq. (49). For the noninteracting QD Hamiltonian, the energy change rate $W_{\mathrm{D}}(t)$ is identified as

$$W_{\mathrm{D}}(t) = \frac{d}{dt}\sum_\sigma \epsilon_\sigma \langle n_\sigma(t)\rangle = \frac{d}{dt}\sum_\sigma \epsilon_\sigma \int \frac{d\omega}{2\pi i} \sum_n \mathcal{G}_\sigma^<(n,\omega)e^{-in\Omega t}, \tag{110}$$

and its Fourier component in the linear regime is found to be

$$W_{\mathrm{D}}(\Omega) = -\sum_\sigma \epsilon_\sigma \int \frac{d\omega}{2\pi}\Omega\mathcal{G}_\sigma^<(1,\omega). \tag{111}$$

Now, by inserting the expressions for $W_D(\Omega)$ and the partial sum (100) into the sum rule (43), one can find that

$$\int \frac{d\omega}{2\pi}\left(\omega + \frac{1}{2}\right)\mathcal{G}_\sigma^<(1,\omega) = -\sum_\ell \frac{\Psi_\ell}{2}(2i\Gamma_\ell)P_{2\sigma}(\Omega). \tag{112}$$

By inserting this integral into the general expression for the heat current, Eq. (37), we obtain the explicit expression for the heat current for the noninteracting case.

$$I_\ell^h(\Omega) = \Upsilon_\ell \hbar \sum_\sigma P_{2\sigma}(\Omega). \tag{113}$$

## E  Interacting case: The Hartree approximation

In the presence of the Coulomb interaction described by the interacting Hamiltonian (61), the Dyson's equation for the QD NEGFs has an additional term proportional to $U$:

$$\begin{aligned}
\widehat{\mathcal{G}}_\sigma(t,t') = \widehat{g}_\sigma(t,t') &+ \int dt'' \frac{U}{\hbar}\widehat{\mathcal{G}}_{d_\sigma, n_{\bar\sigma} d_\sigma}(t,t'')\tau_3 \widehat{g}_\sigma(t'',t') \\
&+ \sum_{\ell \mathbf{k}} \int dt'' \widehat{\mathcal{G}}_{\sigma, \ell \mathbf{k}\sigma}(t,t'')\frac{t_{\ell\mathbf{k}\sigma,\sigma}^*(t'')}{\hbar}\tau_3 \widehat{g}_\sigma(t'',t'),
\end{aligned} \tag{114}$$

where $\widehat{\mathcal{G}}_{d_\sigma, n_{\bar\sigma} d_\sigma}(t,t')$ are the Green's functions between the operators $d_\sigma(t)$ and $n_{\bar\sigma}(t')d_\sigma(t')$. Here we adopt the Hartree approximation (62) so that

$$\mathcal{G}_{d_\sigma, n_{\bar\sigma} d_\sigma}^{R/A}(t,t') \approx \mathcal{G}_\sigma^{R/A}(t,t')\langle n_{\bar\sigma}(t')\rangle. \tag{115}$$

Then, we recover the non-interacting Dyson's equations (103) but with the self energy being now replaced by the Hartree one defined as Eq. (63). The Fourier components of the Hartree self energies in the linear response regime are then

$$\Sigma_{\sigma,\text{HF}}^{R/A}(0,\omega) = \pm i\Gamma + \frac{U}{\hbar}n_{\bar\sigma}(0), \tag{116a}$$

$$\Sigma_{\sigma,\text{HF}}^{R/A}(1,\omega) = \frac{U}{\hbar}n_{\bar\sigma}(1,\Omega), \tag{116b}$$

where $n_{\bar\sigma}(0)$ and $n_{\bar\sigma}(1,\Omega)$ are the $n = 0$ (equilibrium) and $n = 1$ Fourier components of the QD occupation $\langle n_\sigma(t)\rangle$, given by Eqs. (65) and (68), respectively. Hence, the equilibrium retarded/advanced QD Green's functions are modified accordingly [see Eq. (66)] and, according to Eq. (106b), the $n = 1$ components of the retarded/advanced QD Green's functions are now finite [see Eq. (67)].

Since $\mathcal{G}_\sigma^{R/A}(\pm 1, \omega)$ are now finite, the integrals of $\int d\omega\, \mathcal{G}_\sigma^<(1,\omega)$ and $\int d\omega\, \omega\mathcal{G}_\sigma^<(1,\omega)$ will have additional terms. First, from the charge conservation (93), by using the properties of the equilibrium QD Green's functions (66) and the explicit expressions (67) for $\mathcal{G}_\sigma^{R/A}(1,\omega)$, one can get

$$\int \frac{d\omega}{2\pi}\mathcal{G}_\sigma^<(1,\omega) = -\sum_\ell \frac{\Psi_\ell}{2}(2i\Gamma_\ell)P_{1\sigma}(\Omega) + \frac{U}{\hbar}n_{\bar\sigma}(1,\Omega)(2i\Gamma)P_{0\sigma}(\Omega), \tag{117}$$

and this integral, combined with Eq. (67), can be used to obtain the explicit expression for $n_\sigma(1,\Omega)$, resulting in Eq. (69). For the QD Hamiltonian (61), the energy change rate $W_D(t)$ is

identified as

$$W_{\mathrm{D}}(t) = \frac{d}{dt}\left(\sum_\sigma \epsilon_\sigma \langle n_\sigma(t)\rangle + U\langle n_\uparrow(t)n_\downarrow(t)\rangle\right) \approx \frac{d}{dt}\left(\sum_\sigma \epsilon_\sigma \langle n_\sigma(t)\rangle + U\langle n_\uparrow(t)\rangle\langle n_\downarrow(t)\rangle\right),$$
(118)

in the spirit of the Hartree approximation. Its Fourier component in the linear regime is then found to be

$$W_{\mathrm{D}}(\Omega) = -\sum_\sigma \epsilon_{\sigma,\mathrm{HF}} \int \frac{d\omega}{2\pi}\Omega \mathcal{G}_\sigma^<(1,\omega).$$
(119)

Now, by inserting the expressions for $W_{\mathrm{D}}(\Omega)$ and the partial sum (100) into the sum rule (43), one can find that

$$\int \frac{d\omega}{2\pi}\left(\omega + \frac{1}{2}\right)\mathcal{G}_\sigma^<(1,\omega) = -\sum_\ell \frac{\Psi_\ell}{2}(2i\Gamma_\ell)P_{2\sigma}(\Omega) - \frac{U}{\hbar}n_{\bar\sigma}(1,\Omega)\Omega P_{1\sigma}(\Omega)$$
$$- \int \frac{d\omega}{2\pi}\left(\omega + \frac{\Omega}{2}\right)\left(f(\omega)\mathcal{G}_\sigma^R(1,\omega) - f(\omega+\Omega)\mathcal{G}_\sigma^A(1,\omega)\right).$$
(120)

By inserting this integral into the general expression for the heat current, Eq. (37), we can get the explicit expression for the heat current. Finally, the charge and heat currents in the Hartree approximation are obtained as

$$I_\ell^c(\Omega) = \Upsilon_\ell(-e)\sum_\sigma P_{1\sigma}(\Omega) + \Upsilon_\ell'\frac{U}{\hbar}(-e)\sum_\sigma P_{0\sigma}(\Omega)P_{1\bar\sigma}(\Omega),$$
(121a)

$$I_\ell^h(\Omega) = \Upsilon_\ell \hbar \sum_\sigma P_{2\sigma}(\Omega) + \Upsilon_\ell'\frac{U}{\hbar}\hbar \sum_\sigma P_{1\sigma}(\Omega)P_{1\bar\sigma}(\Omega),$$
(121b)

with

$$\Upsilon_\ell' \equiv -\frac{\Psi_\ell}{2}(2i\Gamma_\ell)\Omega(2\Gamma) + \frac{\Psi_\ell - \Psi_{\bar\ell}}{2}(4\Gamma_l\Gamma_r)(i\Omega).$$
(122)

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
