# Peer review of "Heat and charge transport in interacting nanoconductors driven by time-modulated temperatures"

_SciPost Physics, doi:SciPost Phys. 16, 094 (2024)_

## Round 3 · Referee Report · Anonymous (Referee 2) · 2024-2-27

Strengths

Same as in previous report

Weaknesses

Same as in previous report

Report

I read the manuscript and the Author's responses.
I recommend the present version for publication in Scipost Physics.

---

## Round 3 · Referee Report · Anonymous (Referee 1) · 2024-2-27

Strengths

  1. Fully quantum coherent treatment of charge and heat transfer through an interacting quantum dot

Weaknesses

  1. Only a small temperature bias is considered

Report

The authors responded satisfactorily to all the comments I made. Therefore, I can now recommend this manuscript for publication.

Requested changes

no changes

---

## Round 3 · Author Response

Warnings issued while processing user-supplied markup:

  • Inconsistency: Markdown and reStructuredText syntaxes are mixed. Markdown will be used.
    Add "#coerce:reST" or "#coerce:plain" as the first line of your text to force reStructuredText or no markup.
    You may also contact the helpdesk if the formatting is incorrect and you are unable to edit your text.

Dear Editor,

We are grateful for sending us the review reports of our manuscript entitled “Heat and charge transport in interacting nanoconductors driven by time-modulated temperatures” by R. Lopez, P. Simon and M. Lee. The Referees consider that our work presents a number of interesting results that can be verified using modern experiments. We are very happy with their constructive and positive reports that will allow to our work to improve.

We have answered to the Referees' questions/comments. After this exhaustive review, we want to transmit to the Editor that we are very satisfied with the new version that we believe that is now ready for publication as an open access article in SciPost.

Reply to Referee 1

We thank the referee for reviewing our manuscript and for his/her criticism that will help to improve our manuscript. We appreciate very much the comments about our results that are considered interesting and susceptible to being tested in experiments. Please, find below our responses.

  1. If the authors find it useful, can they comment on the possible relationship (or lack thereof) between their approach to time-dependent temperature bias and the Tien-Gordon approach to time-dependent voltage bias?

In the Tien-Gordon approach the effect of a microwave field consists of adding a (spatially constant and) time-dependent electric potential $V_{AC} \cos\omega_0 t$ which is coupled to the charge number operator in leads. It transforms the wave function $\Psi(x)$ to the time-dependent one $\Psi(x,t) = e^{iV_{AC} \cos\omega_0 t} \Psi(x)$, which yields the appearance of quasi-energies $\epsilon \rightarrow \epsilon + n\omega_0$. This approach was initially applied to the tunneling between thin superconducting films through a barrier and revealed that tunneling occurs not only at energy $\epsilon$ but also at the quasi-energy values $\epsilon + n\omega_0$ due to absorption and emission processes of quanta of energy (photons). The current through the barrier is then composed of the sum of all possible absorption/emission processes, each of which is weighted with some probability given by the Bessel functions. This behavior is maintained in the case of quantum dots driven by an ac potential but only in the non-interacting case and in the wide-band limit (when the tunneling broadening is considered constant). When interactions are considered the quantum dot charge needs to be computed self-consistently and the Tien-Gordon description is no longer valid [see Rosa López et al., Low-temperature transport in ac-driven quantum dots in the Kondo regime, Physical Review B 64, 075319 (2001)]. In such a case the current is no longer simply the sum of individual current events evaluated at the quasi-energies weighted by special functions, but much more involved.

The Tien-Gordon approach cannot be directly applied to the time-dependent temperature-driven case which is the main focus of our work. The key difference is that in the Luttinger’s scheme, the time-dependent field is now coupled to the excitation energies as well as the change numbers. Then, the simple analysis based on the quasi-energies is no longer valid: the photo-assisted processes with different n become intermingled with each other. So the current cannot be interpreted in terms of the sum of the processes. Therefore, the Tien-Gordon approach seems not appropriate in our case.

Also, there is one more issue. In the original Tien-Gordon approach, the field is spatially constant. But our study shows that the time-dependent field should not be spatially constant, but instead, a part of the field should be coupled to the barrier as well, in order to predict the correct heat current.

As the referee has recommended we have added a couple of sentences to make clear about the main difference between our formalism and the Tien-Gordon picture [see page 9 in the revised manuscript].

  1. Why in Eq.(24) are the two tunneling amplitudes tlkσ,m and (tlkσ,m')^* calculated at the same times t (not at t and t’)?

Surely the time for (tlkσ,m’)^* should be t’.

We thank the referee for alerting us about this typo. We have corrected it in the revised version.

  1. After Eq.(26), the authors write: "While this value can diverge in the wide-band limit, it is irrelevant in our study..". To me, this saying is a bit confusing: If this term would not be subtracted on the left hand side of Eq.(26), then the right hand side of this equation would diverge. Therefore, this term is relevant in order to obtain physically meaningful finite result. Maybe the authors mean something different.

The relevant physical quantity which is to be measured in experiments is not the energy stored in the leads, but its time derivative, that is, heat current (the change rate of the energy). Hence, the constant, whether it diverges or not, is eliminated once the time derivative is taken. Also, the wide-band limit is a theoretical artifact used to make the results simpler (by eliminating non-essential parts). So, the constant should be finite.

We have stressed better that the term is relevant only for the heat flow in the revised version [see below Eq. (26)].

  1. What is E_{T\ell 0} in the additional unphysical term of Eq.(37) in the linear response? If this is the energy stored in the tunneling barrier, then the additional unphysical term is zero. Since in a linear response, E_{T\ell 0} is calculated at \Psi_{\ell}=0, that is, in the static case. But the energy stored in the tunneling barrier is zero in the static case.

The constant $E_{T\ell0}$, which is not zero, is the energy stored in the barrier for the static case. Under the same reasoning as in the answer for Comment 3, this static value does not contribute to the current. However, this constant $E_{T\ell0}$ also appears in the first-order term of the current. We attribute it to the artifact of our Luttinger’s scheme. We have devoted one paragraph (below Eq. (37)) to the explanation of its appearance and the reason why it can be safely ignored.

In our setup, we apply the field $\Psi(t)$ to the lead, and the additional field $\Psi(t)/2$ to the barrier. So, an effective dynamic energy capacitor is formed, which should not be present in the original system. This effective energy capacitor induces an additional ac heat current between two energy reservoirs, which is the last term in Eq. (37). So, this term should be ignored. We have confirmed this fact by applying our formalism to the non-interacting case. Moreover, the presence of the interaction in the quantum dot does not affect this unphysical term, so the heat current expression, Eq. (37) without the unphysical term is generally correct as long as the linear response regime is considered.

  1. The quantities R, C, and Z have indices that do not match in Figure 2 and in its caption.

The referee is right. We have corrected the indices in Figure 2 to match with those in the main text.

  1. Which equations are used to produce plots shown in Figure 3 ?

The Referee's remark is very useful. Indeed it was not clear from the caption which equations are used to make the lines in Fig. 3. The cross thermal resistance (Fig. 3a) and capacitance (Fig. 3b) are evaluated via Eq. (50) with Y = K, where the cross thermal admittance K is given by Eq. (48b). The cross thermoelectric resistance (Fig. 3c) and capacitance (Fig. 3d) are evaluated via Eq. (50) with Y = L, where the cross thermoelectric admittance L is given by Eq. (48a). In both the cases, the temperatures are finite but the value of the frequency is chosen numerically as small as possible so that the low-frequency limit is taken. For the case of low temperature (solid black lines) there is a comparison with the analytical expressions [Eq. 54(a,b) and 55(a,b)] obtained from the Sommerfeld approach as stated in the caption. We have included this information in the caption of the revised version.

  1. On page 16, the authors write: "It should be noted that the fluctuation-dissipation theorem applied to the heat transport through two-contact systems is no longer valid because scattering events that connect two different terminals induce a nonvanishing term for the equilibrium heat-heat correlation function at the low temperature limit, which is incompatible with the expected behavior of Klr(Ω) [63,64]” Could the authors be more specific by showing an example of what FDT predicts, what they predict, and what the difference is between the two predictions?

Usually, it is assumed that FDT holds for heat transport in which the spectral density of the energy current and the ac heat conductance are related accordingly to

$$ S(\omega) = \hbar \omega T \Re G_{th} \coth \left(\frac{\hbar \omega }{2T}\right) \rightarrow S(0) = 2T^2 G_{th}(0) $$

And this assumption is based on the zero frequency limit for the static case. However, Ref [63] (see Reference List of our work) demonstrated that this is not the case. In the case of heat transport, the heat conductance is not given solely by the energy fluctuations and it has an extra term. Physically, the origin of this extra term can be traced back to finite coupling between the reservoirs, which creates quantum fluctuations of their energy even at $T=0$, when the thermal conductance in the FDT relation vanishes since there are no real excitations that could irreversibly transfer energy between the reservoirs.

To show explicitly the departure of the FDT we need to extend our theory to the calculation of fluctuations which is a very interesting extension for our work in the short future. At this stage, we can solely affirm that our results seem to indicate that FDT cannot hold for ac transport in the heat flow.

Possible misprints: i. After Eq.(8): (In my view, "a" implies any, but "same" is definitely not any)
 with a same frequency -> with the same frequency by a same Fermi distribution -> by the same Fermi distribution

ii. After Eq.(30):
 NEQF -> NEGF

We thank the referee for alerting us about these misprints. We have corrected all of them in the new version.

Reply to Referee 2

We thank the Referee for reviewing our manuscript and for his/her questions/comments that will help to improve our manuscript. Please, find below our responses.

  1. In Ref. 52, a gauge invariant formulation Luttinger’s representation was presented. I think that the authors follow this route, as they Introduce the ‘’gravitational field’’ in the hopping term. I also guess that Eq. (21) results from expanding an exponential after a gauge Transformation. Is that correct? In any case, it would be useful to have more details on the steps from Eq. (8) to Eq. (21).

The formulation in Ref. [52], called as the thermal vector potential theory, rewrites the coupling term between the gravitational field $\Psi(x,t)$ to the energy in terms of the (thermal) vector potential, by using the energy conservation law. One can then find the relation

$$ \partial_t A(x,t) = - \frac{\nabla T}{T}. $$

That is, the spatial variation of the temperature is replaced by the temporal (and spatial) variation of the (thermal) vector potential. The formulation is usually applied to the bulk case, and it is known that this kind of transformation has some pros: the elimination of $T=0$ divergence and the ability to incorporate the magnetic currents.

In our scheme, our gravitational field follows its original definition in the Luttinger's idea

$$ \nabla \Psi(x,t) = - \frac{\nabla T}{T}. $$

That is, our field has nothing to do with the vector potential.

We didn't follow the formulation based on the thermal vector potential theory because we focus on the thermal transport through nanostructures and the dynamic thermal scattering by the nanoconductors such as quantum dots. Then, the original formulation of Luttinger is enough.

  1. In addition and somehow related to the previous item, it is not completely clear how the ‘’gravitational field’’ is related to the temperature bias. In Luttinger’s approach, there is only one field associated to the difference of temperature between the two reservoirs. In Ref. 52, this was substituted by the time-derivative of a vector potential (following the analogy with electromagnetism). In the present paper, I’m not able to find the explicit relation between \psi_\ell and the temperatures. Are two fields necessary instead of a single one? Why? More discussion on these points is most welcome.

From the relation $\nabla \Psi(x,t) = - (\nabla T)/T$, one may regard the gravitational field as the inverse of the temperature (as long as the temperature variation is small enough). Since the temperature is (in a rather macroscopic scale) a function of both the time and the position, the gravitational field can vary with the time and the position. In our scheme, the spatial dependence is denoted by the lead index $\ell$, assuming that the temperature is homogenous in each lead, and we focus on the temporally sinusoidal variation of the temperature.

  1. It would be perhaps interesting and useful to analyze the dc limit, corresponding to zero frequency.

The strictly dc case does not require the Luttinger's scheme because the Landauer-Buttiker formalism works successfully in this case.

In the demonstrations of the non-interacting and the interacting cases (with the Hatree approximation), we indeed studied the low-frequency limit. The resistances, capacitances and RC times shown in Figs. 3, 4, 5 and 6 are the values in the low-frequency limit. It should be noted that the low-frequency limit does not necessarily saturate to the dc case. For example, as shown in Fig. 2, the self-resistance $R_{Y,\ell}$ and self-capacitance $C_{Y,\ell}$ are relevant only in the ac case. It is because the dc current cannot flow through the capacitance. The same reasoning applies to the cross capacitance $C_{Y,lr}$. So, only the low-frequency quantity which is relevant in the dc case is the cross resistance $R_{Y,lr}$. We already have a short discussion on $R_{Y,lr}$ in the first paragraph of Sec. 4.2, which relates the resistances to the density of states in the quantum dot.

  1. Minor details I detected:
  2. There is a missing operator d in Eq. (44)
  3. There is a missing ‘prime’ in the argument of the second t_{lk..} in Eq. (24)

We thank the Referee for alerting us about these misprints. We have corrected all of them in the new version.

---

## Round 3 · List of Changes

1. [At pages 8--9, below Eq. (26)] The sentences are changed to reflect the point (comment 3) raised by the Referee 1.

  2. [At page 9, below Eq. (30)] A new paragraph is added to explain the point (comment 1) raised by the Referee 1.

  3. [Figure 2] The figure is redrawn to have correct mathematical symbols as pointed by the Referee 1 (comment 5).

  4. [At the caption of Fig. 3] A new sentence is added to explain how the plots are produced, as requested by the Referee 1 (comment 6)

  5. Correction of typoes

[Below Eq. (9)] with a same frequency -> with the same frequency by a same Fermi distribution -> by the same Fermi distribution

[At Eq. (24)] a missing ‘prime’ in the argument of the second is added: (t) -> (t')

[Below Eq. (30)] NEQF -> NEGF

[At Eq. (44)] a missing operator d is added

---

## Editorial Decision

published